



# What is Adiabatic Fraction in Cumulus Clouds: High-Resolution Simulations with Passive Tracer

Eshkol Eytan[1], Ilan Koren[1], Orit Altaratz[1], Mark Pinsky[2], and Alexander Khain[2]

[1]Department of Earth and Planetary Science, Weizmann Institute of Science, Rehovot, Israel
[2]Institute of Earth Science, Hebrew University, Jerusalem, Israel

**Correspondence:** Ilan Koren (Ilan.Koren@weizmann.ac.il) and Alexander Khain (Alexander.Khain@mail.huji.ac.il)

**Abstract.** The process of mixing in warm convective clouds and its effects on microphysics, is crucial for an accurate description of cloud fields, weather, and climate. Still, it remains an open question in the field of cloud physics. Adiabatic regions in the cloud could be considered as non-mixed areas and therefore serve as an important reference to mixing. Therefore, the adiabatic fraction (AF) is an important parameter that estimates the mixing level in the cloud in a simple way. Here, we test

different methods of AF calculations using high-resolution (10 m) simulations of isolated warm Cumulus clouds. The calculated AFs are compared with a normalized concentration of a passive tracer, which is a measure of dilution by mixing. This comparison enables us to examine how well the AF parameter can determine mixing effects, and to estimate the accuracy of different approaches used to calculate it. The sensitivity of the calculated AF to the choice of different equations, vertical profiles, cloud base height, and its linearity with height are all tested. Moreover, the use of a detailed spectral bin microphysics

scheme demonstrates that the accuracy of the saturation adjustment assumption depends on aerosol concentration, and leads to an underestimation of AF in pristine environments.

## 1 Introduction

Warm convective clouds were found to have a major role in the high uncertainty that clouds exert on climate change research (Sherwood et al., 2014; Zelinka et al., 2020). Clouds' radiative forcing, defined as the change that anthropogenic aerosols

impose on clouds' radiative properties and life-cycle (e.g. the aerosols indirect effect), is considered to be negative (i.e. cooling; IPCC 2013; Boucher et al. 2013). On the other hand, the feedbacks of warm clouds on the changing climatic system were recently shown to be positive, due to reduction in cloud cover (Ceppi et al., 2017; Nuijens and Siebesma, 2019). A major drawback in understanding the effects of shallow convection on climate and their representation in models are the processes of entrainment and mixing. These processes have a major impact on cloud properties and hence on their radiative forcing and

feedbacks. As an example, high aerosol loading conditions increase the number of droplets and their surface area to volume ratio, which increases the diffusion efficiency. This increases the liquid water content in the core of the cloud (Albrecht, 1989) and the evaporation at the cloud's edge. Thus, the intensity of mixing plays an important role in the non-monotonic response of clouds to aerosol loading (Small et al., 2009; Dagan et al., 2017). Mixing also affects convection and its vertical fluxes, which are important for climate models (De Rooy et al., 2013). Mixing effects on microphysical cloud properties is still an open





question in cloud physics (Khain and Pinsky, 2018). Additionally, the occurrence and location of adiabatic regions in shallow clouds are still under debate (Gerber, 2000; Khain et al., 2019). This has significant consequences for shallow clouds, since cloud top height, as well as microphysical cloud properties, depend on the existence/absence of an adiabatic core. Further, obtainment of the adiabaticity level is important for parameterizations of the vertical mass fluxes (De Rooy et al., 2013), and for remote sensing retrievals, in which the radiation transfer calculations depend on adiabatic microphysical profiles (Merk

et al., 2016). Hence, the usage of a simple parameter that characterizes the mixing level can be very beneficial. The ideal way to evaluate dilution by mixing is to use a passive tracer, which is a conservative variable in moist adiabatic processes, (i.e. does not change during evaporation/condensation). Sub-cloud tracers are preferable over natural conservative variables, such as total water mixing ratio or equivalent potential temperature, as they are absent from the clouds' surroundings. This eliminates the need for knowledge about the conservative variable's initial profile and assumptions on its mixing processes.

However, such fictitious tracers do not exist in in-situ measurements and remote sensing and are only being used in numerical simulations, aiming for process-level understanding of mixing. The level of adiabaticity (i.e. deviation from a perfect adiabatic state) can also be a measure of mixing in cases where radiation and sedimentation are negligible, thus it is widely common to use adiabatic fraction ($AF$) as a proxy for adiabaticity.

The $AF$ is determined as:

$$AF = \frac{LWC}{LWC_{ad}} \tag{1}$$

where LWC is the liquid water content ($g/m^3$) at a specific location, and $LWC_{ad}$ is the theoretical liquid water content that a parcel would have if it was lifted adiabatically from the cloud base to a specific height. The definition of $LWC_{ad}$ is not consistent in the literature; many studies define $LWC_{ad}$ using the moist adiabatic lapse rate as derived by Yau and Rogers (1996), with inherent saturation adjustments assumption (i.e., $S(t, z) \approx 0$). This definition considers $LWC_{ad}$ as the maximal potential

of LWC. This maximal value does not describe the true potential because it ignores the fact that the potential of $LWC_{ad}$ is limited by the condensation efficiency. Saturation adjustment assumes that the total amount of water vapor that exceeds the concentration for saturation will condense instantaneously. Such assumption ignores the relaxation time for condensation that determines the condensation efficiency, and depends on the available surface area of the droplets. Cases of clouds with high supersaturation values can occur in clouds with low droplet concentrations (i.e. low surface area to volume ratio) or very strong

updrafts. The various approaches for $AF$ calculations differ in the way by which they calculate the reference $LWC_{ad}$. The values of $LWC_{ad}$ can be obtained using parcel modeling or direct calculations, which can be performed in a bulk approach (e.g. using conservation of energy or water mass of all phases (Brenguier, 1991), or by using analytical thermodynamic considerations (Khain and Pinsky, 2018; Pontikis, 1996). The different methods are detailed in section 2.3.

AF is commonly used to study the effects of mixing on clouds' microphysical structure. Observations (Freud et al., 2008) and

numerical modeling (Zhang et al., 2011) used AF to show the effects of mixing on the effective radius profile in cloud fields. Conditioning aircraft measurements of cumulus and stratiform clouds according to $AF$ were used to examine the effects of mixing on the width of the droplets size distribution (DSD; Pawlowska et al. 2006; Pandithurai et al. 2012; Kim et al. 2008; Bera 2021. $AF$ is also commonly used in mixing diagram analyses for determination of mixing types (Gerber et al., 2008;





Schmeissner et al., 2015). Additionally, it is used to calibrate in-situ aircraft measurements (Brenguier et al., 2013). Some

studies approximated $AF$ by normalizing LWC by the maximal measured value at a given height ($LWC_{max}$; Bera 2021).
However, some clouds may not contain adiabatic regions, or their adiabatic pockets may not be sampled, thus, the normaliza-
tion of in-situ measurement data by the maximal value might lead to an overestimation of $AF$, as $LWC_{max} \leq LWC_{ad}$. In
this study, we evaluated the accuracy of different methods and assumptions for $AF$ estimation, by simulating several single
warm cumulus clouds in high-resolution (10 m), using different aerosol concentrations. The dynamic model is coupled to a

spectral bin microphysics model for explicit representation of the microphysical processes and the resulted supersaturation
field. The high resolution allows to solve the turbulent fluxes in more detail, and reduces the model dependence on sub-grid
parameterizations, which improves mixing representation. Moreover, the small grid spacing enables better detection of local
maxima in the 3D field (e.g. LWC, supersaturation, updraft). We confront $AF$ with the sub-cloud layer tracer that represents
the level of dilution by mixing.

Details about the model and the tracer are provided in sections 2.1 and 2.2, respectively. In section 2.3 we derive analytical
equations for $LWC_{ad}$, and present the different assumptions that can be made for its calculations. In the results section, we
compare three different $AF$ calculation methods (equations) to the sub-cloud layer tracer (3.1). In section 3.2 we show the
effects of different assumptions on the accuracy of $AF$ calculation, and in 3.3 we quantitate the accuracy of the various as-
sumptions by sampling numerous cloudy points in space and along the clouds' lifetime, providing a larger dataset for statistics.

## 2    Methods

### 2.1    Model description

The clouds were simulated using the System for Atmospheric Modelling (SAM; Khairoutdinov and Randall 2003, http://
rossby.msrc.sunysb.edu/~marat/SAM.html) coupled with the Hebrew University Spectral Bin Microphysical scheme (SBM;
Khain et al. 2004; Fan et al. 2009). SAM is a non-hydrostatic, inelastic model with cyclic boundary conditions in the horizontal

direction. Sub-grid turbulence parameterization was performed using a 1.5-closure scheme. Analysis by Pinsky et al. (2021)
shows that turbulent motions in this design obey the $-\frac{5}{3}$ law. To avoid the effects of the cloud on itself via the cyclic boundaries,
we chose the domain size to be 5.12 km, which is much larger than the cloud scale ($\sim$800 m diameter). The horizontal resolution
was set to 10 m, and the vertical resolution to 10 m up to 3 km, and 50 m for the last kilometer (maximal cloud top is 2 km).
The time resolution was 0.5 seconds. Initial vertical profiles of water vapor mixing ratio and potential temperature (inversion

at 1500-2000 m), and constant large-scale forcing and surface fluxes were taken from the BOMEX case study (Siebesma
et al., 2003). The horizontal background wind was set to zero and aerosols were distributed only below cloud base (600 m).
The cloud was simulated for one hour, and was initialized by a perturbation of 0.1 K in the center of the domain, with a
horizontal radius of 500 m and a vertical radius of 100 m (from the surface). The perturbation decays to zero as a Cosine
square function of $0 < x < \frac{\pi}{2}$ from the center to the edge of the radius, and random noise is added (perturbation type 7 in SAM

manual). The SBM is based on solving kinetic equations for size distribution functions of water drops and aerosol. Both size
distributions are defined on a doubling mass grid containing 33 bins. The drops radii range between 2 $\mu$m and 3.2 mm. The




size of aerosols serving as cloud condensational nuclei (CCN) ranges between 0.005 and 2 $\mu$m. Following Jaenicke (1988) and Altaratz et al. (2008), the size distribution of the aerosols was represented by a sum of three log-normal distributions describing fine, accumulation, and coarse mode aerosols, typical for the maritime boundary layer. Three clouds with different

aerosol concentrations (Na) were simulated, with Na= 5, 50, and 500 $cm^{-3}$.

## 2.2  Passive tracer setup

For quantification of the dilution level of the cloud, we used a passive tracer that disperses in space and time by advection and turbulent diffusion that is set according to the sub-grid scheme. The tracer is uniformly distributed in the sub-cloud layer, from the surface up to 600 m (mean cloud base). Throughout the simulation, the measured concentration is normalized by the sub

cloud initial concentration; therefore, a concentration equal to unity indicates no dilution. Fig. A1 in the appendix shows three snapshots: the tracer's initial spatial distribution, its distribution and values at the time of the cloud's maximal development (33 minutes), and at the end of the simulation, after 55 minutes.

## 2.3  Adiabatic fraction calculations

Although AF was used in many studies over the years, there are different methods for calculation of $LWC_{ad}$, which are often

not well defined in the literature (details of the calculations are often missing). The value of $LWC_{ad}$ can be calculated in different ways that differ by method, assumptions, and practical implementations. In this section, we present three commonly used methods for AF calculation and the following assumptions that can be made.

The equation for supersaturation (S) for an adiabatically ascending parcel is given by Korolev and Mazin (2003):

$$\frac{1}{(S+1)}\frac{dS}{dt} = A_1 w - A_2 \frac{dLWC}{dt}, \quad i.e., \quad \frac{dlog(S+1)}{dt} = A_1 w - A_2 \frac{dLWC}{dt} \tag{2}$$

$$A_1 = \frac{g}{T}\left(\frac{L_w}{c_p R_v T} - \frac{1}{R_a}\right) \tag{2a}$$

$$A_2 = \frac{1}{\rho_v} + \frac{L_w^2}{c_p R_v T^2 \rho_d} \tag{2b}$$

where w is the updraft velocity and $A_1$ and $A_2$ are thermodynamics parameters, which depend on temperature and water vapor mixing ratio that vary with altitude.

Eq. 2 is obtained by differentiating $S = \frac{(e-e_s)}{e_s}$ , and using a quasi-hydrostatic approximation that is valid for updrafts weaker

than 10 $\frac{m}{s}$. e is the water vapor partial pressure, $e_s$ is the saturated water vapor partial pressure over liquid, g is the gravity acceleration, T is temperature, $L_w$ is the latent heat of evaporation, $c_p$ is the heat capacity of air under constant pressure, $R_v$ and $R_a$ are the gas constants of water vapor and dry air, respectively, and $\rho_v$ and $\rho_d$ are the density of water vapor and dry air, respectively.

The first term in the RHS of Eq. 2 is the source of S by adiabatic cooling, and the second term is the sink of S due to water

vapor loss and latent heat release by condensation. Since Eq. 2 does not include effects of mixing, the value of LWC equals





$LWC_{ad}$. Considering the changes in S in the vertical direction only, and transforming the time domain to vertical coordinates using w leads to:

$$w\frac{dlog(S+1)}{dz} = A_1 w - wA_2\frac{dLWC_{ad}}{dz} \tag{3}$$

And the LWC in an adiabatic parcel is:

$$LWC_{ad}(z) = \int_0^z \frac{A_1(z')}{A_2(z')}dz' - \int_0^z \frac{1}{A_2(z')}\frac{dlog(S+1)}{dz'}dz' \tag{4}$$

where z=0 at cloud base.

One can see that $LWC_{ad}$ is not only a function of z, but also depends on temperature and humidity (via parameters $A_1$ and
$A_2$), as well as on vertical velocity and aerosols through the supersaturation term. When $S << 1$, Eq. 4 can be simplified to:

$$LWC_{ad}(z) = \int_0^z \frac{A_1(z')}{A_2(z')}dz' - \int_0^z \frac{1}{A_2(z')}\frac{dS}{dz'}dz' \tag{5}$$

Eq. 5 shows that in regions where S increases with height (e.g. near cloud base, or in pristine environments), $LWC_{ad}$ will
be smaller than its maximal value, because some amount of water vapor in excess of supersaturation remains in the gas phase.
At the exception of these cases, the S term is small compared to the first term on the RHS of Eq. 5. Neglecting the term which
includes the supersaturation, we can write:

$$LWC_{ad}(z) \approx \int_0^z \frac{A_1(z')}{A_2(z')}dz' \tag{6}$$

Taking $\frac{A_1}{A_2}$ as a constant leads to the well-known linear $LWC_{ad}$ profile, which is the first assumption to be examined in this
study (section 3.2.1).

Two alternative approaches can be used to calculate $LWC_{ad}$. One is using the total water-mixing ratio $q_t = q_l + q_v\left(\frac{g}{kg}\right)$,
which is a conservative value in moist adiabatic processes. $q_v$ is the water vapor mixing ratio and $q_l$ is the liquid water mixing
ratio. At cloud base $q_l = q_{l_0} \approx 0$, and so $q_{t_0} = q_{v_0}$. For undiluted parcels $q_{t_0} = q_t(z)$, and at any altitude above cloud base
$q_t(z) = q_l(z) + q_v(z)$. Assuming saturation adjustment (i.e. $S(z) = 0$) means that $q_v = q_{vs}$, where $q_{vs}$ is the water vapor mixing
ratio in saturation that can be calculated according to the Clausius-Clapeyron equation. $LWC_{ad}$ can then be defined using $q_l$,
as:

$$LWC_{ad}(z) = [q_{vs_0} - q_{vs}(z)]\rho_d(z) \tag{7}$$





Such an approach was used by Gerber et al. (2008) (personal communication).

The third approach is to use the conservation of moist static energy ($h$), where $h = L_w q_l + c_p T + gz$. Differentiating $h$ with respect to z, conserving it with height ($\frac{dh}{dz} = 0$) and multiplying by $\rho_d$ gives (Schmeissner et al., 2015):

$$LWC_{ad}(z) = \frac{c_p}{L_w} \int\limits_0^z \rho_d(z') \left( \frac{g}{c_p} + \frac{dT}{dz'} \right) dz' \tag{8}$$

Eq. 8 shows that the difference between the lapse rate of an adiabatic parcel and the dry adiabatic lapse rate ($\Gamma_d = \frac{g}{c_p}$) is due to condensation, and thus can be translated into $LWC_{ad}$. We note here that this method avoids the use of saturation adjustments.

## 3 Results and Discussion

### 3.1 Comparison between the three methods

In this work, we solved the Lagrangian equations presented above from the outputs of the Eulerian model, using the assumption that in a time scale of $\sim 5$ minutes the thermodynamic profiles in the cloud are fixed during growth and mature stages. It implies that the profiles of temperature and humidity can be used to predict the conditions to which a parcel in the cloud base would be exposed as it ascends. The most accurate way to consider profiles of temperature (T(z)) and specific humidity ($q_v(z)$) for $LWC_{ad}$ calculations is to obtain them from the undiluted core of the cloud, where $q_v$ is maximal and T is warmer due to release
of latent heat. If there is a perfect undiluted adiabatic core, its AF value is equal to one, and it will coincide with the maximum normalized value of the tracer (Tr). Fig. 1 shows cross-sections of Tr and the three different methods for calculating AF (see list below) when it reached its maximal height and mass (33 minutes). The sensitivity of the methods to the choice of profiles is tested in Fig. 1 by calculating each method twice, first with accurate "least diluted" profiles, and second, by approximating the adiabatic (undiluted) profiles, using the points with the highest updraft values at each level.
The methods are denoted as:

1. $AF_{ref}$: calculated according to Eq. 6 using the in-cloud profiles of $A_1$ and $A_2$. This method for AF calculation will be used as the reference method from herein (reasoning for this choice is provided below).

2. $AF_{dTdz}$: calculated according to Eq. 8.

3. $AF_{qt}$: calculated according to Eq. 7.

The accurate estimations of the adiabatic vertical profiles of T and $q_v$ were obtained here by averaging the values of those parameters in the voxels containing the highest 1% Tr values at each altitude, and the results are presented in Fig. 1a-c. The cross-section of Tr is provided in Fig. 1d.

The vertical profiles of T and $q_v$ that were used in Fig. 1a-c (based on the simulated Tr) can also be calculated using the maximal values of LWC or updraft. It is hard to obtain these types of profiles from in-situ measurements that do not contain



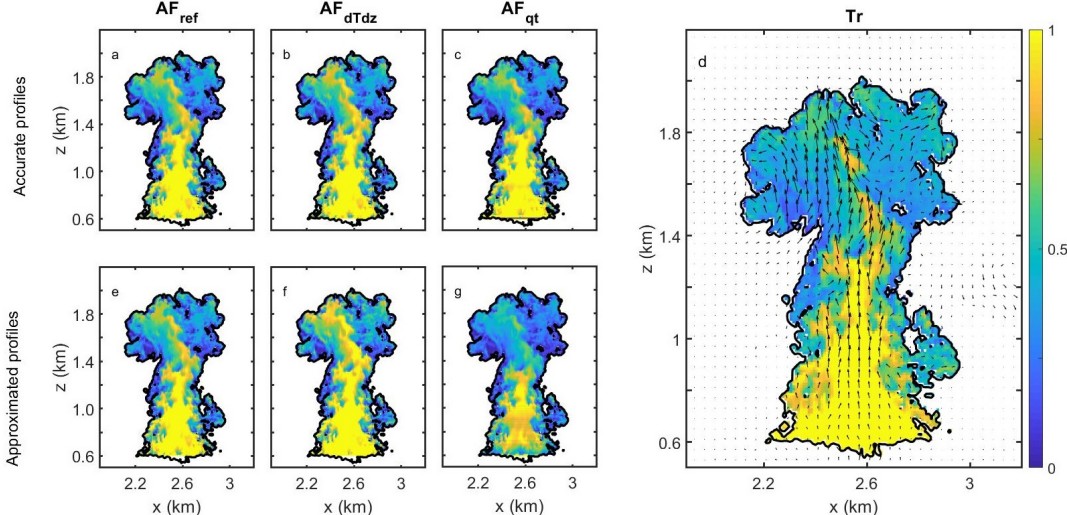

**Figure 1. Cross-sections of the tracer and the three AF methods.** Presented for a cloud with a 500 $cm^{-3}$ aerosol concentration at the time of its maximal top height. The AFs in the upper row were calculated using accurate adiabatic profiles of T and $q_v$. Profiles in the bottom row were approximated using high updrafts. (a) The method for calculating AF used as reference ($AF_{ref}$): by solving Eq. 6. (b) Same as panel a, for $AF_{dTdz}$; using Eq. 8 (c) Same as panel a, for $AF_{qt}$; using Eq. 7. (d) Normalized concentration of the sub-cloud layer tracer (Tr). (e) Same as panel a, for approximated profiles. (f) Same as panel b, using approximated profiles. (g) Same as panel c, using approximated profiles.

the theoretical tracer, nor the full 3D distribution of the cloud variables. Thus, for the sake of simplicity, and in order for the methods presented in this paper to be comparable to measurements, we approximated the profiles by averaging the values in the voxels with the highest 5% updraft values at each altitude. This methodology was used to estimate the T and $q_v$ profiles throughout this study. It is shown that $AF_{ref}$ remains almost similar when using either the approximated or accurate profiles. On the other hand, $AF_{qt}$ and $AF_{dTdz}$ exhibit some underestimations and overestimations compared to the accurate profiles,

respectively. These differences are explained in detail below. Fig. 2 presents the differences between each AF method when using the approximated profiles and the Tr values (as shown in Fig. 1d). The apparently good agreement between $AF_{ref}$ and Tr, as presented in Fig. 1e is more closely examined in Fig. 2a, where differences are detected. Close to cloud base, the AFs experience non-realistic, non-homogenous values due to the inhomogeneity of the cloud base. Moreover, the values of LWC and $LWC_{ad}$ near the cloud base are small, hence their ratio exhibits high sensitivity even when differences from the reference

are minor (chosen according to the highest updraft). These differences are no longer observed around 100 m above cloud base. For the sake of comparison with Tr, the points near cloud base with AF>1 were set to one. Determination of the cloud base height was achieved using the vertical profile of the cloud horizontal cross-section area. All clouds exhibited a local maximum in the cross-section area around 600 m during their growing stage. Aiming to choose the cloud base as a level that can represent the cloud with "enough" cloudy voxels, we chose to define it as the height above the level of initial condensation, in which





the area covers 90% of the local maximal area. Changing the criteria threshold from 90% to 33% can decrease the cloud base
height by up to 30 m. This definition, using the 90% criteria, was found to be stable for all simulations during the growing stage
of the clouds, and is considered optimal for AF calculations since it maintains an optimal agreement between $AF_{ref}$ and Tr in
the regions of high values. When comparing the cross-sections of Tr with each AF presented in Fig. 1 (more than 100 m above
cloud base), one can see that the AFs decrease toward the cloud edge faster than Tr (Tr>AF; see Fig. 2a). This is because AF

is also affected by evaporation, and not only dilution, as in the case of Tr (i.e. mechanical mixing). The opposite is observed in
higher levels, at slightly diluted regions, where $Tr < AF_{ref} < 1$. These regions represent a more complex difference between
AF and Tr, which is also caused by condensation and evaporation. Tr can change only due to mechanical mixing and hence,
is almost a one-directional process; once the parcel is diluted, it has low probability to restore its initial Tr concentration.
This means that Tr has a memory of the mixing history, unlike AF that has a source/sink process. A parcel can regain liquid

water after a mixing event, if supersaturation is reached again at a later time. This means that a parcel in the margins of the
cloud can be diluted, decreasing both Tr and LWC (AF), but later, if the parcel gains vertical velocity and supersaturation,
it can condense water and compensate for some of the LWC loss (keeping Tr the same, while increasing AF). Fig. 2a shows
that the blue regions of AF>Tr are voxels of relatively strong updrafts that are part of the flow pattern of the toroidal vortex
(for an elaborate discussion of the vortex see Zhao and Austin 2005). Using the velocity arrows, the regions of AF>Tr can be

back-trajected to regions where the toroidal vortex entrains environmental air. Those parcels that mix with entrained air are first
diluted, but then the flow pattern of the toroidal vortex inside the cloud forces them upward; Hence, they cool and re-condense
some of the water they lost. The phenomenon of rapid growth of droplets in an updraft following an entrainment event was
suggested as a mechanism for rain initiation (Baker et al., 1980; Yang et al., 2016). Correlation of the blue regions (where
AF>Tr) with strong updrafts (as part of the toroidal vortex) was found for different time-steps and different cloud simulations.

Fig. 1f shows the cross-section of $AF_{dTdz}$ values, and Fig. 2b shows its difference from Tr. Here, a very good agreement
with $AF_{ref}$ is observed, although $AF_{dTdz}$ with the approximated profiles is slightly larger in the sub-adiabatic regions at
higher levels (i.e. having smaller values of $LWC_{ad}(z)$). This is explained by the fact that $AF_{dTdz}$ considers the difference
between $\frac{dT}{dz}$ and the dry lapse rate as a consequence of condensation, and uses it to define $LWC_{ad}$. Diluted parcels are colder
than the adiabatic core because they were mixed with colder environmental air, and may have experienced evaporation. This

difference between absolutely adiabatic and slightly diluted parcels increases with height, as the parcel is aging. For these
reasons, using diluted voxels to estimate the adiabatic profiles will lead to a larger temperature gradient (more negative) that
is closer to the dry lapse rate, falsely inferring less condensation, and biasing $LWC_{ad}$ toward smaller values. The arguments
above explain the difference between Fig. 1a-b where $AF_{dTdz} \approx AF_{ref}$, and Fig. 1e-f, where $AF_{dTdz}>AF_{ref}$. These findings
suggest that $AF_{ref}$ is less sensitive to the choice of adiabatic profiles, because it is constrained by two free parameters (T and

$q_v$), rather than only T. The final method, $AF_{qt}$, is shown to be very sensitive to the choice of profiles, as presented in Figs. 1c,g.
The deviation from Tr in Fig. 2c demonstrates a substantial underestimation of $AF_{qt}$, due to a similar argument as discussed
above for $AF_{dTdz}$. Using a slightly diluted parcel, with a smaller $q_v$ or $q_{vs}$ compared to the core, is expected to falsely infer
more condensation and larger $LWC_{ad}$. The bias is stronger for this method because it depends on $q_v$ (or its estimation as
$q_{vs}(T)$). The simplicity of this method is also its downfall; since $q_v$ is an order of magnitude larger than ql (liquid mixing





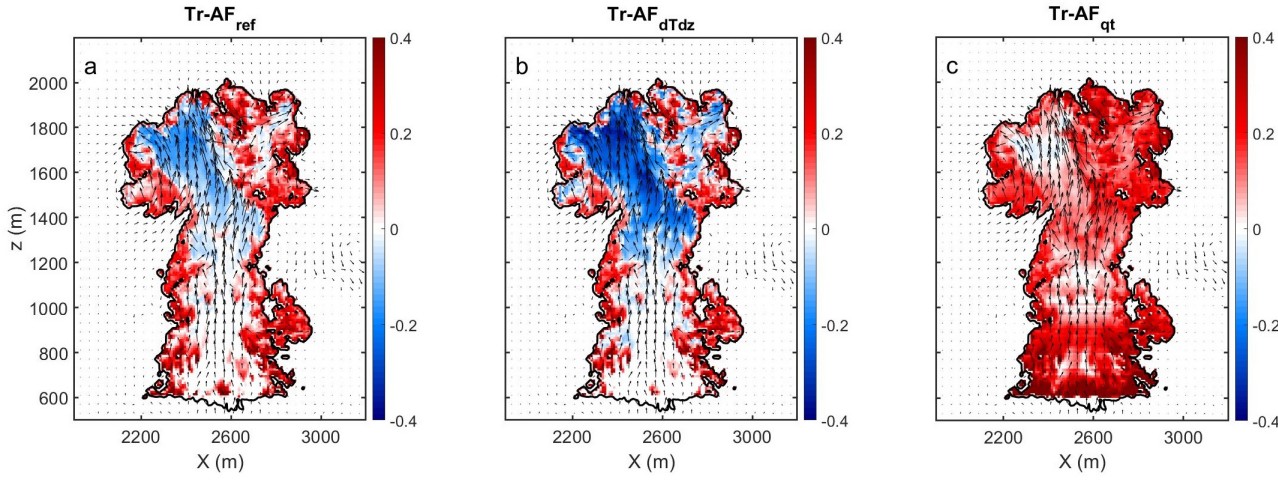

**Figure 2. Cross-sections of the differences between the various AF methods and the tracer**. Vertical cross-sections for the differences between methods using approximated profiles (Fig. 1e-g) and the tracer (Fig. 1d). (a) Difference between AFref and Tr (Fig. 1e minus Fig. 1d). (b) Same as a, for $AF_{dtdz}$. (c) Same as a, for $AF_{qt}$.

ratio), small errors can cause significant effects when estimating $LWC_{ad}$ using only $q_v$. Another disadvantage of $AF_{qt}$ is that it is commonly used with the saturation adjustments assumption (i.e. $S \approx 0$), by estimating $q_v$ to be $q_{vs}$. This assumption can lead to underestimation of $AF_{qt}$ in conditions of pristine environment (low aerosol concentrations) as explained in detail in section 3.2.3. The results of this section suggest that the analytical solution for $AF_{ref}$ using Eq. 5 is a more accurate and stable method to calculate AF, as it shows similarity to Tr and robustness for different choices of vertical profiles. We also note that

profiles of T and $q_v$ are often obtained from the environment (see section 3.2.2). This will lead to substantial errors when using $AF_{dTdz}$ or $AF_{qt}$, which showed sensitivity to the choice of profiles. Furthermore, Eq. 5 allows isolating different assumptions such as linearity, with height and saturation adjustments.

## 3.2   Testing the effects of assumptions made when calculating AF

Next, we examine the effects of several commonly used assumptions on $AF_{ref}$ calculation, in order to estimate their impact.

Although the magnitude of the difference that should be considered as significant depends on the application, we define here a considerable difference as 0.1, which is 10% of the maximal $LWC_{ad}$.

The approaches that will be examined next are:

1. $AF_{linear}$: Using Eq. 6 and keeping $\frac{A1}{A2}$ constant from the cloud base.

2. $AF_{env}$: Using the sounding (environmental) profiles in Eq. 6.

3. $AF_s$: Including the supersaturation term using Eq. 5.



4. $AF_{+50}$: Using Eq. 6, but estimating the cloud base height to be 50 m higher.

5. $AF_{-50}$: Using Eq. 6, but estimating the cloud base height to be 50 m lower.

**Figure 3. The effects of assumptions on AF.** Vertical cross-sections of the differences between the various assumptions used when calculating AF. (a) The method of calculating AF that is used as the reference ($AF_{ref}$; as presented in Fig. 1e) (b) $AF_{ref}$ subtracted from an AF that is linear from cloud base ($AF_{linear}$). (c) $AF_{ref}$ subtracted from $AF_{env}$, calculated using sounding (environmental) profiles. (d) $AF_{ref}$ subtracted from $AF_s$, which considers the supersaturation term (Eq. 5). (e) The error in AF caused by overestimating cloud base by +50 m (f) Same as e, with a -50 m error.





### 3.2.1 Linear $LWC_{ad}$

$AF_{linear}$ is a very common method, based on the assumption that $LWC_{ad}$ is linear with height (i.e. neglecting the dependence
of $\frac{A1}{A2}$ on temperature and humidity). This implies that $\frac{A1}{A2}$ can be used as a constant, based on the known values at the cloud
base (Pontikis, 1996). Indeed, small negative differences from to the non-linear $AF_{ref}$ (i.e., underestimation) are observed in
Fig. 3b when using $\frac{A1}{A2}$ as a constant from the cloud base. Pontikis (1996) derived such a solution for stratocumulus clouds, and
noted that the error using this method would increase for deeper clouds. On the same note, Brenguier (1991) argued that the
linear assumption is valid for shallow clouds (depth of up to 200 hPa, $\approx 2km$). This emphasizes that the usage of the $AF_{linear}$
method is restricted to shallow clouds, and should be used with care or avoided altogether for deeper clouds.

The changes in the growth rate of $LWC_{ad}$ ($\frac{A1}{A2}$ in Eq. 6) with height, which can lead to deviation from $AF_{linear}$ (or $AF_{env}$,
discussed next), occur mostly due to changes in $A_2$. This is because $A_1$ (Eq. 2a), which is a parameter in the term for cooling
by ascent, depends only on temperature, and exhibits a negligible change in the case of our shallow clouds. $A_2$ (Eq. 2b), which
relates to the S sink term (by condensation), depends on T and $q_v$, and increases with height. Fig. 4 demonstrates the sensitivity
of $A_2$ to $q_v$ and T, by presenting the differences in the profiles of $A_2$ from the true profile (calculated using the original $q_v$ and
T profiles), and when keeping T or $q_v$ as a constant (with the value set at cloud base). The true profile of $A_2$ (in blue) shows
an increase with height, as mentioned earlier. One can see that the $A_2$ profile with a constant T (red curve) is very similar to
the true profile. On the other hand, the profile's gradient decreases significantly when using $q_v$ as a constant (yellow line). This
demonstrates that depletion of water vapor in higher levels of the cloud is the major factor that impacts $A_2$ values (see the
inverse relation to water vapor mixing ratio in Eq. 2b) and the deviation of $LWC_{ad}$ from its linear relations.

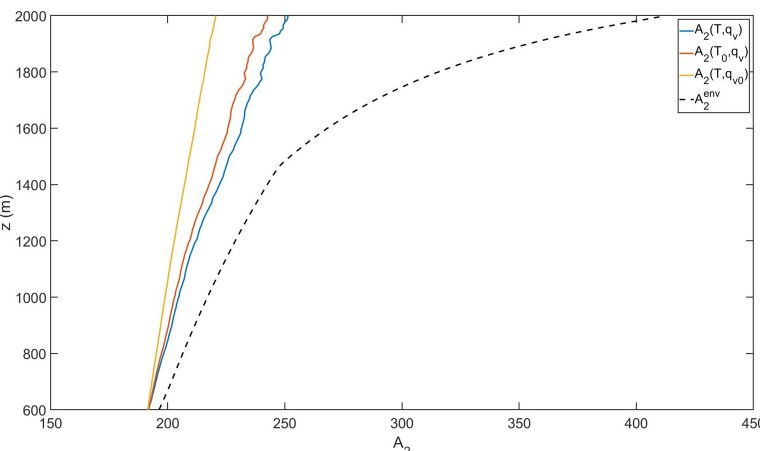

**Figure 4. The sensitivity of $A_2$ profile to temperature and humidity**. The growth rate of $LWC_{ad}$ depends on the ratio of $\frac{A1}{A2}$ (see Eq.
6). $LWC_{ad}$ changes with height depend mostly on $A_2$, since $A_1$ exhibits little sensitivity. The blue curve is the true profile of $A_2$. Red and
yellow $A_2$ profiles are calculated using constant temperature and humidity, respectively, and the dashed black line is the profile obtained
using the environmental sounding.



### 3.2.2 $LWC_{ad}$ using sounding profiles (environmental profiles)

The advantage of using environmental profiles is that they can be obtained from sounding data, and can be considered as constant reference values for an ensemble of clouds (the whole cloud field). Such application can have large errors in cases where the T and $q_v$ profiles in the cloud's core and the environment exhibit large differences (e.g. penetration of cloud into the inversion layer, or into higher levels of the atmosphere). The profile of $A_2$ when calculated using the environmental profiles is given in dashed black line in Fig. 4. It shows that the environmental $A_2$ profile values are larger than the profiles in the core of the cloud (especially in the inversion layer above 1500 m, where $q_v$ decreases fast). The larger $A_2$ values lead to a smaller $LWC_{ad}$ ($A_2$ is in the denominator in Eq. 6), and hence, to an overestimation of $LWC_{env}$, as seen in Fig. 3c. The small overestimation witnessed in our case (trade wind cumulus in Barbados), will probably be greater for deeper clouds, where the gradients between the core and the environment are larger.

### 3.2.3 The role of the supersaturation term in conditions of low aerosol concentrations

Considering the profiles of T and $q_v$ is necessary if one wishes to dismiss the saturation adjustment assumption (i.e. $\frac{dS}{dz} \approx 0$). This assumption is almost inherent in most previous works that we know of. The supersaturation can be significantly greater than zero in regions of high updrafts and/or small droplet concentrations (for example, in the first tens of meters above cloud base, and in pristine environments). Thus, if one wants to achieve accuracy near cloud base, or compare different clouds under different aerosol loading conditions (e.g. studying aerosol effects on cloud mixing), one has to use $AF_s$ as calculated by Eq. 5. The second term in Eq. 5, referred to here as the S term, depends on the vertical profile of S. It is worth noting that S profiles are available only in modeling studies, since in-situ measurements of S cannot reach the desired accuracy, as far as we know. Fig. 3d demonstrates that for cases where the cloud develops under conditions of high aerosol concentrations (Na=500 $cm^{-3}$), neglecting the S term introduces a negligible underestimation of AF near the cloud base ($AF_s > AF_{ref}$). However, this is not the case for cleaner environments (lower Na). Neglecting the S term means assuming that the parcel condenses all of the excess water vapor that forms as it ascends and cools. This overlooks the limited condensation efficiency in pristine environments that prevents a full consumption of water vapor by the drops, and lets S increase (i.e., $\frac{dS}{dz}$ and the second term are larger than zero). To evaluate this effect, we simulated two additional clouds in a cleaner environment, with lower aerosol concentrations (Na=50 $cm^{-3}$ and 5 $cm^{-3}$). Before analyzing these simulations, we first had to make sure that there is no significant sedimentation in the clouds at this stage. Sedimentation creates liquid water loss and downdrafts, which violate the adiabatic assumption, and lead to a deviation of AF from Tr. For the example examined in this section, we used Tr to assure that sedimentation can be neglected in the time-steps we chose for comparison (timing of maximal cloud top height of each cloud). Fig. A2, in the appendix, shows vertical cross-sections of $AF_{ref}$ and Tr for the cloud simulations with Na=50 $cm^{-3}$ and 5 $cm^{-3}$, at the time steps used for this example. There is a good agreement between $AF_{ref}$ and Tr for Na=50 $cm^{-3}$ at 33 minutes, and Na=5 $cm^{-3}$ at 31 minutes. This is not the case for Na=5 $cm^{-3}$ after 40 minutes, when the cloud precipitates, and sedimentation can no longer be neglected. In Fig. A2 we observe regions in the clouds with large differences between Tr and $AF_{ref}$ values.

In Fig. 5, we present the deviation of $AF_s$ (when including the saturation term) from $AF_{ref}$ for the three different simulations.



Fig. 5a (same as Fig. 3d) shows that the deviation is negligible for high Na. The deviation increases and spreads to higher

levels when Na is smaller (Fig. 5b-c). Under pristine conditions (Na=5 $cm^{-3}$, Fig. 5c), the underestimation of $AF_{ref}$ spreads

throughout the entire cloud column. The profiles of S are depicted in Fig. 5d for all cases, presented as the mean S of the

voxels with the highest 5% updraft in each level. Fig. 5d shows that saturation adjustment is a reasonable approximation for

AF calculations in polluted clouds, because the S profile is smaller, and more importantly- almost constant. The gradient of

the S profile of the Na=50 $cm^{-3}$ case is positive also far from the cloud base (at 1000-1400 m). It doesn't introduce very

large errors to $AF_{ref}$ (as can be seen in Fig. 5b) because the gradient is relatively small, and thus the second term in Eq. 5 is

negligible compared to the first term. Near cloud base, the first term in Eq. 5 is small, and thus, even a relatively small gradient

in S can be significant, and lead to a large difference between $AF_{ref}$ and $AF_s$.

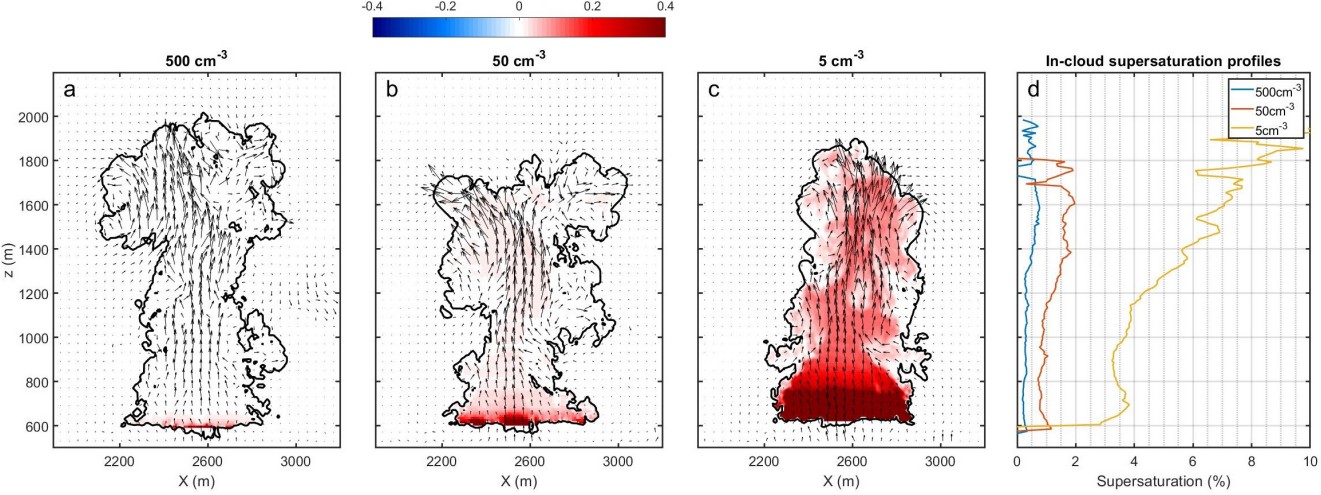

**Figure 5. Microphysical effects on AF**. The droplet concentration (i.e. aerosols) affects AF through the profile of supersaturation inside the cloud. (a)-(c) present cross-sections of $AF_s$ (with the supersaturation term) minus $AF_{ref}$. (a) For high Na of 500 $cm^{-3}$ (as in Fig. 3d). (b) For 50 $cm^{-3}$ at 33 minutes. (c) For 5 $cm^{-3}$ at 31 minutes. (d) In-cloud supersaturation profiles.

### 3.2.4 AF sensitivity to cloud base heights

Last, we tested how sensitive AF calculations are to the errors in the estimation of the cloud base height. Fig. 3e-f shows the

deviation from $AF_{ref}$ when having an error of ±50 m in cloud base height. When overestimating (underestimating) cloud

base height, as in Fig. 3e (3f), the estimated $LWC_{ad}$ is smaller (larger), and AF is larger (smaller). These results demonstrate

that such small errors in a parameter that is often taken for granted can introduce large errors in adiabaticity estimation. For

example, the Lifting Condensation Level (LCL), which is often used to approximate cloud base height, can be obtained from

a tephigram, or calculated by several proposed analytical equations. Calculating LCL from surface conditions, as suggested

by Bolton (1980), Lawrence (2005), or Romps (2017), approximates cloud base height to be 515, 550, or 525 m, respectively.





These approximations are lower than the height that was found as optimal for AF calculations using the sub-cloud layer tracer ($\approx$600 m, depending on simulation time and cloud properties). Additionally, LCL is known to be an underestimation of cloud base height when the convective parcel is driven by perturbation in temperature. In such a case, the perturbation reduces the parcels' relative humidity, and therefore the parcel starts the condensation at a higher altitude (above the LCL).

This can cause an overestimation of $LWC_{ad}$ and an underestimation of AF. The opposite will occur when the convection is driven by a humidity fluctuation (Hirsch et al., 2017). We note that most in-situ measurements and space-borne remote sensing observations lack the tools required to measure the cloud base height, and usually estimate it based on LCL.

### 3.3   Mean differences between the assumptions with time

So far, we examined the AF calculations at one time-step the time of maximum development of the clouds ($\sim$33 min). The

robustness of the results tested by estimating the deviation of each method from $AF_{ref}$ over height, and as a function of time, along the cloud's lifetime. Since the deviations are more pronounced in regions of high AF, we chose to consider for this analysis only the cloudy regions with $AF_{ref} > 0.5$. Note that these sub-adiabatic regions are important, and highly debated. Fig. 6 shows the mean deviation for a cloud with Na=500 $cm^{-3}$. Here, we observe that the statistics along the cloud's lifetime agree with the instantaneous qualitative pictures presented in Fig. 3. As demonstrated in Fig. 6a, the linear assumption ($AF_{linear}$)

underestimates AF for altitudes above cloud base. $AF_{env}$, using the environmental profiles (Fig. 6b), exhibits a small over-estimation, which becomes significant near the cloud top, at the inversion layer. Here, we define a significant difference as larger than 0.1 in absolute value (marked by the black contour). Considering the S term in the polluted case does not have a considerable effect (Fig. 6c). Errors in the estimations of cloud base by 50 m lead to relatively large errors in AF- up to 1000 m (Fig. 6d-e). Repeating the same analysis for the clean case (with Na=5 $cm^{-3}$) gives similar results for all methods but $AF_s$ (see

Fig. 7c) which supports the argument made in section 3.2.3. The time series is shorter in this case because sedimentation starts after 33 minutes. It seems that the observed differences between the AF calculation methods don't change with time during the growth stage of a particular cloud, for both pristine and polluted conditions.



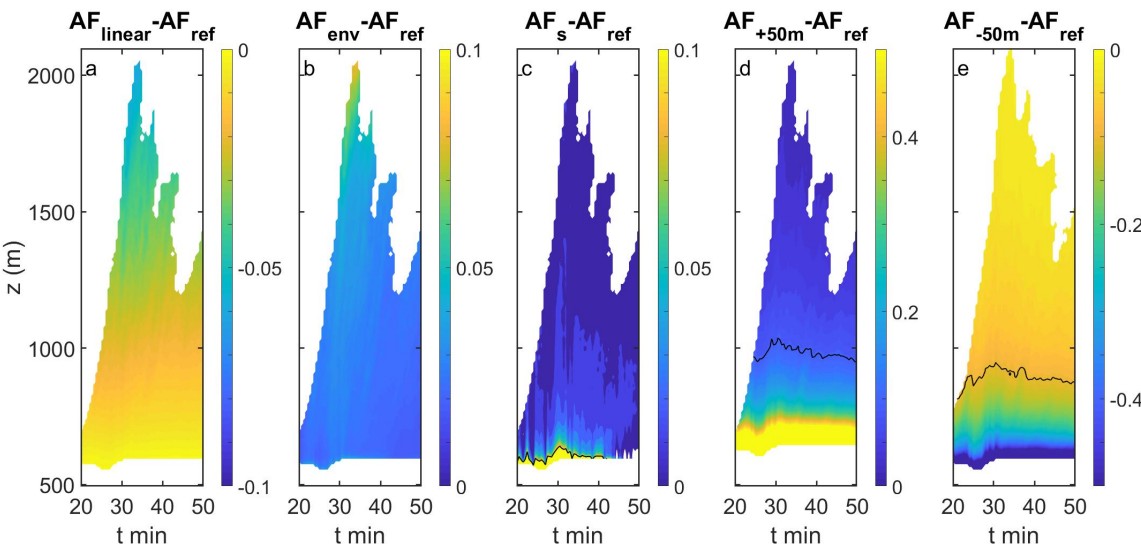

**Figure 6. Mean differences in AFs vs time and altitude for a polluted cloud**. The horizontal average of the difference of each assumption from $AF_{ref}$ for regions with $AF_{ref}>0.5$. (a) The difference between the linear method $AF_{linear}$ and $AF_{ref}$. (b) Same as a, for $AF_{env}$ (c) Same as a, for $AF_s$ (d) The error in AF produced by an error of +50 m in cloud base height. (e) Same as d, for -50 m. Black contour marks the 0.1 or -0.1 deviation. Note the different scales for the different panels.

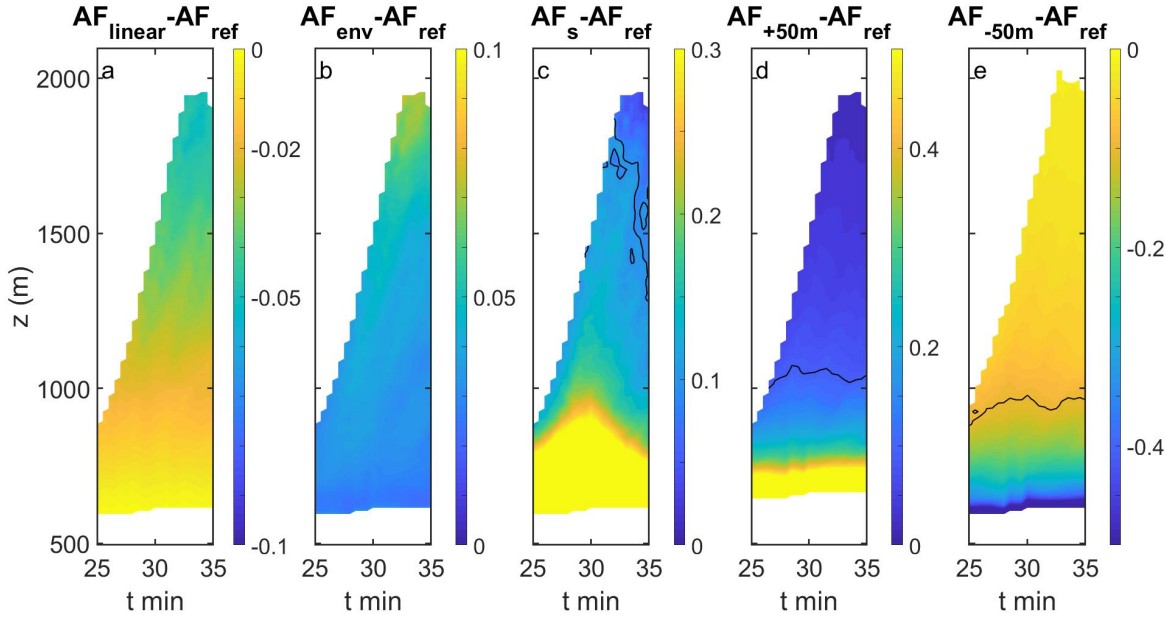

**Figure 7. Mean differences in AFs vs time and height for a clean cloud.**





## 4    Conclusions

An accurate calculation of the adiabatic fraction (AF) is crucial in two main aspects:

1. An accurate AF can promote a high-resolution measure of the mixing state of sampled parcels. This may advance the research of mixing processes in shallow clouds and their effects, which remain open questions in the field of cloud physics.

2. To Allow mapping of the occurrence and extent of adiabatic regions in shallow clouds, which are still under debate. Adiabatic processes are simpler to predict, hence adiabaticity is assumed in remote sensing retrieval algorithms and
340        cloud parametrizations in weather and climate models.

Answering these questions can improve our process-level understanding, in-situ measurements, and remote sensing retrievals. This will improve models' representation of shallow convection, and may reduce the magnitude of the shallow clouds' contribution to the uncertainty in weather and climate models.

In this study, we tested the accuracy of different approaches that are commonly used to calculate adiabatic fraction (AF). We
used high-resolution (10 m) simulations of isolated trade wind cumulus clouds, that solve the turbulent flow down to scales that are rarely achieved. This enables a better representation of mixing, and relaxes the dependency on sub-grid parameterization schemes. A sub-cloud layer's passive tracer (Tr), which is an accurate measure of mechanical mixing, was added to the simulations and was used as a reference point. The calculation of AF using Eq. 4-6 (which is based on a Lagrangian model) from a snapshot of an Eulerian model, demanded bridging the gap between these two different perspectives. This was achieved
by assuming stationarity of the thermodynamic profiles in the cloud core. Through the validation of AF calculations by Tr values, we found that this assumption holds for the temperature profile, but not for the specific humidity ($q_v$). The method that is based only on $q_v$ (Eq. 7) exhibited a weaker agreement with Tr. Some regions in the cloud emphasized the important differences between AF and Tr. While Tr follows the complex flow in the cloud and records all mixing events, AF is based on a one-dimensional model whose reference lies in the core. For that reason, AF cannot describe processes that occur in the
margins. Moreover, condensation that occurs after a mixing event can delete records of earlier evaporation/dilution events. As an example, the toroidal vortex drives entrainment events followed by updrafts, which cause some parcels to experience dilution and evaporation (decrease in Tr and AF), followed by condensation that increases AF.

Three different calculation methods of AF (Eq. 5,7,8) were compared with the tracer. The most robust method (Eq. 5) is an analytical solution that allows the isolation of different assumptions and evaluation of their accuracy. The important findings
and their implications are as follows:

Assuming a linear profile of $LWC_{ad}$, or using the sounding profiles of temperature and humidity instead of the in-cloud profile, produces small errors at higher levels of shallow clouds ($\sim 2$ km). The small error in AF for shallow clouds obtained using environmental profiles suggests that it can be used as a constant reference for all clouds in the field.

The saturation adjustment assumption was integrated into the calculation of AF in most previous studies. Testing this assump-
tion on clouds that develop in different environmental conditions (with different aerosol concentrations), revealed that it can





lead to underestimation of AF. A simulation of a cloud in a pristine environment (Na=5 $cm^{-3}$) yielded high supersaturation values (compared to the polluted case), and led to an underestimation of AF when assuming saturation adjustment (i.e. $\frac{dS}{dz} \approx 0$). This means that comparing clouds' mixing under different aerosol loading when using the saturation adjustment assumption may neglect some of the microphysical effects on clouds' dynamics, and mixing in particular.

AF was found to be sensitive to errors of $\pm 50$ m in the estimated cloud base height; especially in the first few hundred meters above cloud base. Determining cloud base height is challenging for aircraft in-situ measurements, and is often obtained by estimating the lifting condensation level (LCL) from a tephigram or analytical solutions. Three analytical solutions that were tested here (Bolton, 1980; Lawrence, 2005; Romps, 2017) differed by 45 m, and underestimated the cloud base height that was optimal for AF calculations. Underestimation of the cloud base height can lead to a larger $LWC_{ad}$, and thus to the underes-

timation of AF. This can lead to an underestimation of the extent of adiabatic regions in shallow clouds. Accurate estimation of AF near the cloud base is challenging because these levels include a ratio of two small numbers (LWC and $LWC_{ad}$), and are not homogenous as mostly assumed. Moreover, the calculation of $LWC_{ad}$ in these levels exhibit high sensitivity to the determination of cloud base and the representative supersaturation profile.

All simulations demonstrated the existence of an adiabatic core (i.e., high values that are close to one for both AF and Tr) up

until the cloud top. While the core is wide at the lower parts of the cloud, it narrows and breaks down to smaller fragments at higher levels. The extent and frequency of adiabatic parcels in different levels of the cloud will be assessed in a subsequent study.

Finally, we point out the limitations of using AF in clouds deeper than the shallow convective clouds of the boundary layer:

1. AF is based on a quasi-hydrostatic equation, which is valid for updrafts smaller than $10 \frac{m}{s}$.

2. Supersaturation in clouds with strong updrafts can increase, leading to underestimations of AF when the S term is neglected.

3. Sedimentation of particles from higher levels of deep clouds can increase LWC in their lower levels and lead to an overestimation of AF.

4. The rate of change in $LWC_{ad}$ is dominated by the parameter $A_2$, which changes as water vapor is depleted in clouds,

meaning that $LWC_{ad}$ ceases to be linear. The large differences expected in deep clouds between the in-cloud and environmental profiles suggest that the latter is prone to large biases when used to predict AF.





## Appendix A

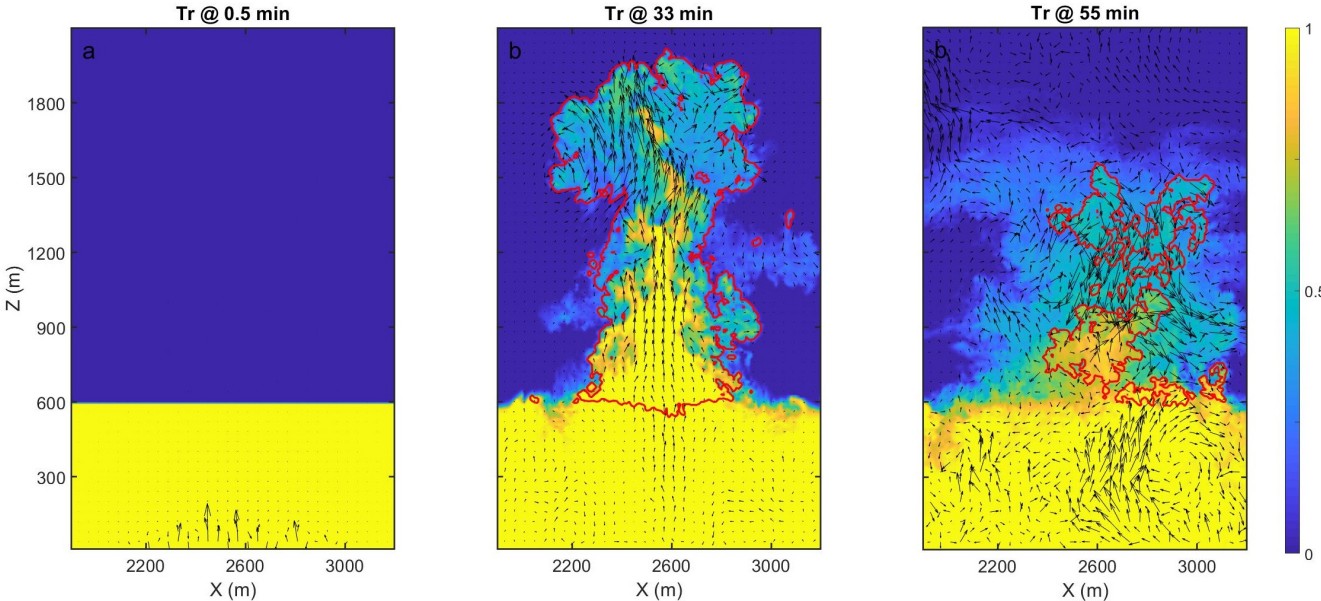

**Figure A1. Sub-cloud layer's passive tracer.** Vertical cross-sections of the tracer along the X-axis in the middle of the domain. Tr is the tracer's mixing ratio normalized by its initial mixing ratio in the sub-cloud layer. (a) The initial distribution at the beginning of the simulation. (b) Distribution at the time of the cloud's maximal development (33 minutes). Red contours mark the cloud boundaries where the liquid mixing ratio is smaller than 0.01 g/kg. (c) Same as b, for the end of the simulation, when the cloud is nearly completely dissipated.

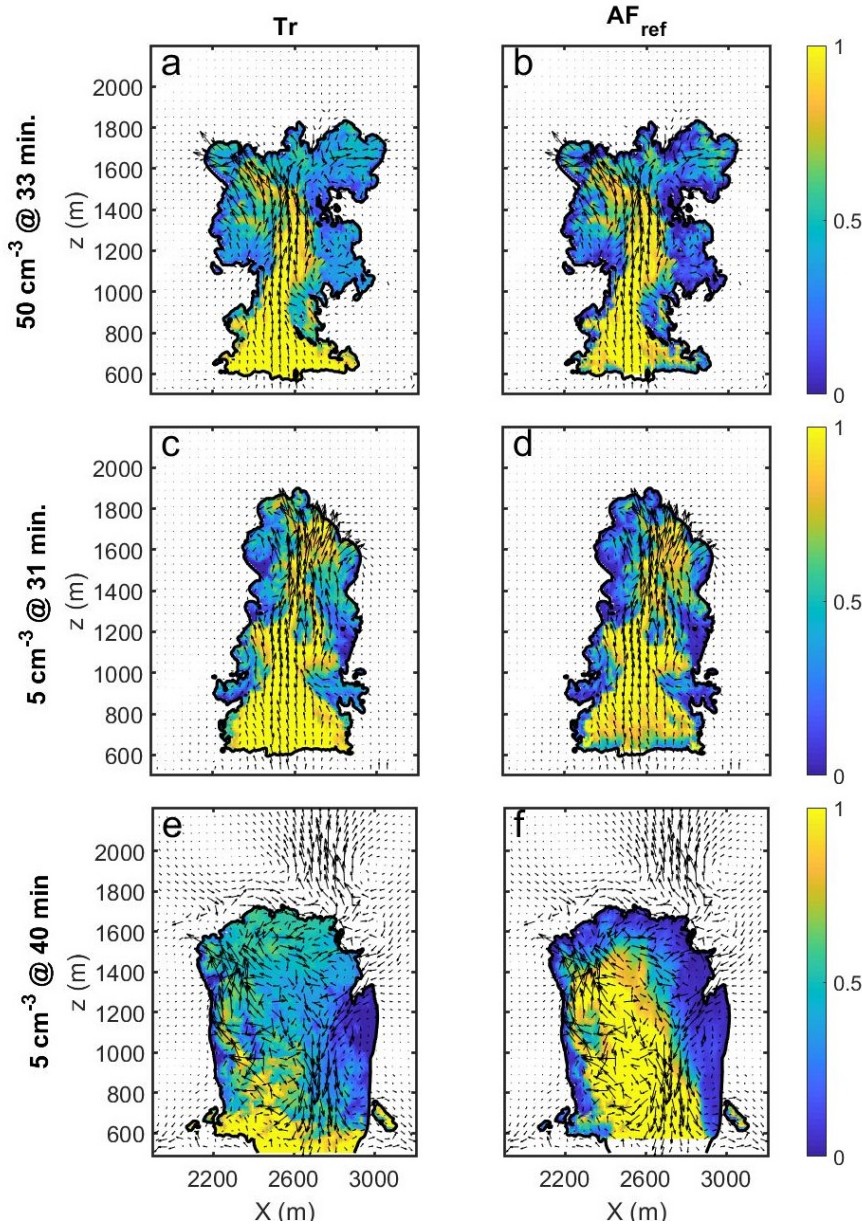

**Figure A2. Comparison of** $AF_{ref}$ **and Tr for low Na**. Cross-sections of $AF_{ref}$ and Tr for lower Na simulations. (a) Tr for 50 $cm^{-3}$, at 33 min. (b) Same as a, for $AF_{ref}$. (c) Tr for 5 $cm^{-3}$ at 31 min, prior to intense sedimentation (b) Same as c, for $AF_{ref}$. (e) Tr for 5 $cm^{-3}$ at 40 min during intense sedimentation. (f) Same as e, for $AF_{ref}$.



*Code availability.* The SAM codes should be requested from Prof. Marat Khairoutdinov from the School of Marine and Atmospheric Sciences, Stony Brook University.

*Data availability.* The microphysical and thermodynamical profiles used to initialize the simulation can be obtained upon request from the corresponding author.

*Author contributions.* E.E., I.K. and A.K. jointly conceived the principal idea. E.E. carried out the analysis. E.E., I.K., O.A., A.K. and M.P. discussed results and wrote the paper.

*Competing interests.* The authors declare that they have no conflict of interest.

*Acknowledgements.* This project has received funding from the European Research Council (ERC) under the European Union's Horizon 2020 research and innovation programme (CloudCT, grant agreement No 810370). A. Khain and M. Pinsky were supported by the grants from the Department of Energy (DE-964SC0008811) and by the Israel Science Foundation (Grants 2027/17 and 2635/20). E. Eytan was supported by the Weizmann Institute Sustainability and Energy Research Initiative





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
