# Peer review of "Revisiting Adiabatic Fraction Estimations in Cumulus Clouds: High-Resolution Simulations with Passive Tracer"

_Atmospheric Chemistry and Physics, 2021_

## Referee Comment (RC2)

Review report of "What is adiabatic fraction in cumulus clouds: high-resolution simulations with passive tracer" by Eytan et al. 2021

**General comments**:

This manuscript compares various method in calculating adiabatic fraction in cumulus clouds, which is an import parameter in quantifying the mixing level. The authors compared three ways in calculating the adiabatic LWC and their impacts on the AF. Besides, the authors examine the assumptions made in previous studies about the calculation of AF. This manuscript is clear about their method but lacks more explanations in the application of this study. I recommend a revision to emphasize the significance of this study more in either introduction or the discussion paragraph. Other specific comments and the minor corrections are below.

**Specific comments**:

- This study mentioned that it aims at identifying the errors from the assumption in calculation of AF in observations, but the manuscript only shows the results from an idealized model study. Can the author provide more clear linkage that how the results can improve the analysis of observation data?

- The reference AF is based on Equation 6 without considering the supersaturation (Line 166). The Section 3.2 shows that not considering supersaturation leads to errors especially in the lower cloud region of cleaner scenario. I think it better to use the full equation (Equation 5) to calculate the LWC_ref since it has less assumption.

- What are the recommendations for the calculation of AF in the future studies?

- This study has removed large-scale wind. How are the results influenced by a mean wind?

**Minor corrections:**

- Figure 2: it is better to show the differences by the AF-Tr, which is more straightforward for the readers to tell overestimation or underestimation.

- Figure 6 and Figure 7: it is better to use red-white-blue colors to show the differences (similar color scheme as Figure 3b-3f)

- Line 224: $q_l$ should have the l as subscript?

---

## Author Comment (AC1)

**Replay to the reviewers**

**"What is adiabatic fraction in cumulus clouds: high-resolution simulations with passive tracer" by Eytan et al. 2021**

We would like to thank the reviewers for their comments that helped us to improve and clarify the manuscript.

To address all the comments and remarks, the manuscript has been revised. We summarize below the novelties and the main modifications performed in the manuscript to clarify and strengthen the results:

1. The significance and applicability of the results are now emphasized and clearly stated.

2. The abstract and introduction were revised to emphasize that this is a theoretical study that compares different approaches that were used previously to calculate AF, and revisits some of their assumptions.

3. The summary and conclusions section was revised such that the take home message for calculating AF (e.g. list of points to consider) are summarized clearly.

4. The results are explained in a more detailed way now with additional information. For example; the effect of the toroidal vortex on the differences between AF and a conservable scalar.

- Point by point responses are presented below (in blue).

**Summary:**

In this paper, the authors assess various methods used for estimating adiabatic fraction (AF) in a non-precipitating cumulus cloud. A High-resolution LES model with bin microphysics was used for simulating the cloud field. The AF computed using different methods was compared against the AF calculated using a passive tracer.

I am not convinced about the key results and conclusions that are drawn from this study. Additional details must be provided to understand the results and to assess the significance of this work.

**Answer:** Thank you for the detailed review. Additional details and explanations were added to the manuscript in order to address all the comments. Please see all the details in the answers below.

**Major comments:**

1.  In section 2.3, the LWCad calculation in Eq. 5, 7 and 8 assume that the parcel under consideration is adiabatic. So, I am wondering how these equations can be used for calculating the adiabatic LWC from a cloud field that is affected by entrainment/mixing. The temperature and water vapor fields from the simulation will be affected by entrainment/mixing. So, using these fields in Eqs. 5, 7 and 8 would violate the assumptions used in deriving these expressions. Thus, the LWCad computed using the technique mentioned in this study would not be correct.

    **Answer:** As the reviewer rightly mentioned, using an adiabatic parcel model to estimate $LWC_{ad}$ has its limitations since not all the parts of the simulated cloud (the higher levels for example) contain pure adiabatic parcels that can be used to calculate the adiabatic profiles. Hence, this issue was the first thing we addressed in the paper. In section 3.1 we show sensitivity tests of each method of $LWC_{ad}$ calculation to the chosen profiles. Fig. 1 in the paper shows that the method we chose to use for the rest of the paper ($AF_{ref}$ that is based on eq. 5,6) do not show significant sensitivity to the chosen profiles, partly because it is using two variables (specific humidity and temperature) that their mixing effects compensate one another. This is not the case in the other methods that are either using humidity ($AF_{qt}$, Eq. 7; in which mixing causes overestimation), or temperature ($AF_{dtdz}$, Eq. 8; in which mixing causes underestimation). In fig. A2 below (added to the revised appendix of the paper as fig. A2), we show the calculated $LWC_{ad}$ profiles by the three methods: when using different subsets of voxels that were chosen by several thresholds on the passive tracer concentration (i.e. representing regions in the cloud in different dilution levels). Indeed, the figure shows that there are no pure undiluted parcels (represented by a threshold of *Tr>0.99*) above the inversion (curve of black dots), but, using the slightly diluted parcels (tracer>0.7) to estimate $LWC_{ad}$ is not introducing large errors in our reference method ($AF_{ref}$). We note that the lowest threshold value that was used for the profiles presented in Fig. 1a-c was a tracer concentration of 0.67 in the highest levels

of the simulation. This is not the case for the other methods ($AF_{qt}$, $AF_{dtdz}$; fig. A2b,c) that clearly show overestimation or underestimation of $LWC_{ad}$. The method that is based on a temperature profile ($AF_{dtdz}$) show smaller biases because the relative change in temperature between pure core and the environment is small (1-2 degrees change over a magnitude of ~290 K). This point is tightly related to comment No. 6 that suggests using the linear assumption with cloud base properties (as derived by Rogers and Yau, 1996). We would like to note that this derivation (that is tested in section 3.2.1) is using a strong assumption about the temperature and humidity profiles instead of estimating them, (i.e., that $A_1/A_2$ is constant with height). One of the goals of this paper is to revisit such assumptions and test their robustness. In the revised manuscript, we added figure A2 to the appendix of the paper and we rewrote parts of section 3.1 to make it clearer.

Line 171: *"The accurate estimations of the adiabatic vertical profiles of T and $q_v$ were obtained here by averaging the values of those parameters in the voxels containing the highest 1% Tr values at each altitude (minimal threshold that was used was Tr=0.67 in the high levels of the cloud), and the results are presented in Fig. 1a-c. The cross-section of Tr is provided in Fig. 1d."*

Line 181: *"It is shown that $AF_{ref}$ remains almost similar when using either the approximated or accurate profiles. On the other hand, $AF_{qt}$ and $AF_{dTdz}$ exhibit some underestimation and overestimation compared to the accurate profiles, respectively. These differences are explained in details below and sensitivity tests to the chosen profiles according to different thresholds on Tr values is presented in the appendix for all three methods (Fig. A2)."*

[Figure]

***Figure A2. LWC_ad profiles of different approaches for different estimations of adiabatic profiles.*** *LWC_ad(z) was calculated according to eq. 6 (**a**), eq. 8 (**b**) and eq. 7 (**c**). Taken from a snapshot of a cloud with aerosols concentration of 500 cm^-3 at the time of maximal development (33 minutes). The temperature and humidity profiles were used by averaging all points of each layer according to a certain threshold on sub-layer tracer normalized concentration (Tr). Black dots are for nearly pure undiluted parcels with Tr>0.99, green dashed line for nearly adiabatic (Tr>0.9), and red and blue curves include also slightly (Tr>0.7) and strongly diluted (Tr>0.1) parcels. It is shown that there are no pure adiabatic parcels above the inversion. Nevertheless, the use of slightly diluted parcels (with Tr>0.7) in our chosen reference method do not introduce large biases to LWC_ad and AF accordingly.*

2. The AFs computed using the methods in section 2.3 are compared against the AF computed using the passive tracer. Is it a fair comparison to compare AFs calculated using two very different variables? The passive tracer is a conserved variable whereas LWC is not. Both these variables to some extent can be used for determining the adiabatic core, but once mixing occurs, then a one-to-one comparison may not be fair. Can the authors comment on that? If the authors agree, then what is the significance of the observations and conclusions from the current study?

**Answer:** We thank the reviewer for this comment. First, we note that in the absence of evaporation/re-condensation after a mixing event, AF can be considered as a conservative variable, similar to a passive scalar. Hence, even though the tracer and AF have their differences, they still share common properties (especially in volumes with relative humidity > 100%). For this reason, the tracer (which is an accurate measure of dilution) is used as a first order approximation for AF. Several assumptions were taken to obtain AF, which is based on a Lagrangian model (with its inherent assumption; some were mentioned by the reviewer earlier), from an Eulerian model outputs (as discussed above and in section 2.3). For this reason, we used the tracer to test the robustness of our AF calculations. This issue is discussed in lines 378-381:*"A sub-cloud layer's passive*

*tracer (Tr), which is an accurate measure of mechanical mixing, was added to the simulations and used as a reference point. This model configuration enabled to better control AF and the complex processes that it represents, and to give a theoretical framework that allowed testing the accuracy of different approaches that are commonly used to calculate adiabatic fraction (AF)."*

We note that in most regions of the cloud the relative humidity is above 100%, which make the AF comparable to the tracer. In the next comment we discuss and suggest more complex reasons that act to deviate the two measures one from another.

Line 183 the text was revised to emphasize that the tracer is used only as a first order approximation: *"If there is a perfect undiluted adiabatic core, its AF value is equal to one, and it will coincide with the maximum normalized value of the tracer (Tr),thus Tr can be used as a first order approximation for AF."*

As the reviewer mentioned, in the core of the cloud the two measures should have a good agreement. In figure R1 below, we show the cross section of (AF-Tr) with the core ($Tr > 0.9$) marked by a black contour and magenta contour that marks the $Tr = 0.8$ line. One can see that the difference between the two is small in those regions.

[Figure]

**Figure. R1.** Vertical cross section of $AF_{ref}$-Tr at the upper parts of the polluted cloud ($N_a$=500 cm$^{-3}$) at the time of maximal development (33 min.). Black contour marks the core where Tr>0.9 and magenta contour is for Tr=0.8.

3. In Figure 2(a), AFref>AF_scalar in the upper half of the cloud (the blue-colored region). The discussion related to this (lines 190-210) attributed it to the presence of toroidal vortices and enhanced updraft. The evidence provided to support this conclusion is not very concrete. The issues raised in the previous two points are relevant here. AFref>AF_scalar, this could also be due to a lower estimated LWC_ad. Since the calculations are based on the simulated cloud fields, the adiabatic LWC obtained using Eqs 5, 7 and 8 would be an underestimation compared to the actual LWCad as the cloud field is affected by entrainment. This is evident from the passive tracer field in Fig. 1(d). Thus, AFref> actual AF, and the actual AF would be very close to the AF estimated using the passive tracer.

**Answer:** We appreciate this comment that helped us clarify this issue in the revised manuscript. The issue about the credibility of our calculation of AF is addressed above in answer number 1. Figure A2 (presented in answer 1) and lines 211-223 explain that the bias of AF will not necessarily be underestimation, depending on the method that is used. Since we assume that our calculation of the reference method (AF$_{ref}$) gives a good estimation and is not prone to large biases due to the use of in-cloud profiles, we would turn to clarify our explanation of the regions where Tr<AF$_{ref}$<1. First, we note that if the reason was due to underestimation of the LWC$_{ad}$ profiles, AF would be larger in the core as well (in regions of AF≈1), but there we see that AF≈Tr (see fig. 2a and 1d, and figure R1 above). Moreover, while the passive tracer is a simple conservable variable that flows according to the velocity fields, AF is based on a 1D parcel model. For this reason, unlike the tracer, it cannot consider horizontal motions and processes that occur away from the core (accompanied by mixing) and cause deviation from the core profiles. Such an example is given in the text and concerns a different condensation rate after a mixing event. If a parcel is highly diluted it remains with very small droplets concentration, then, if it continues to rise, local high supersaturation can occur. This local phenomenon cannot be considered in AF calculations because it is not represented in the estimated cloud core profiles. Thus, this parcel that experiences secondary nucleation (that is not assumed in the parcel model) and higher condensation rates than expected by the adiabatic parcel model. This will lead to local regions with $\frac{dLWC}{dz} > \frac{dLWC_{ad}}{dz}$ which means that AF ($\frac{LWC}{LWC_{ad(z)}}$) will unexpectedly increase with height. This hypothesized unique mechanism is tightly related to an old idea suggested by Baker et al., (1980) that explains super-adiabatic droplets [Baker, M.B., Corbin, R.G. and Latham, J., 1980. The influence of entrainment on the evolution of cloud droplet spectra: I. A model of inhomogeneous mixing. *Quarterly Journal of the Royal Meteorological Society*, 106(449), pp.581-598.]. Here we show that the toroidal vortex can continuously generate such conditions near the cloud's top. Figure R2 below shows a close-up on the upper part of the cloud, presenting the difference between the tracer and AF (**a**). In panel (**b**) we show the supersaturation field with its local high values that are found in ascending parcels just above the entrainment region of the toroidal vortex (on the left side of the cloud). The green contour aims to bound the "red regions" (where Tr > AF and the magenta contour marks regions of tracer=0.8 from the figure above (comment No. 2). Panel (**c**) shows the

supersaturation values and points on correlations between high S values on the upper left side of the cloud and low droplets concentration. Finally, panel (**d**) presents $AF_{ref}$ and show that the region bounded by the green contour contain intermediate values of ~0.8 that smoothly decrease to ~0.6. These values could be a consequence of increased AF from the toroidal vortex (as discussed above) or dissipated and diffused remnants of core fragments. Our results suggest the first explanation. This interesting role of the toroidal vortex is not the topic of this study and will be further studied in the future. In this paper it is only given as an example of regions in the cloud where the 1D adiabatic model can deviate from the tracer. For the sake of clarity of this manuscript, we re-wrote lines 221-240:

*"The opposite is observed in higher levels, at slightly diluted regions, where Tr< $AF_{ref\_}$<1. These regions represent a more complex difference between AF and Tr, which is also caused by condensation/evaporation. Tr can change only due to mechanical mixing and hence, is almost a one-directional process; once the parcel is diluted, it has low probability to restore its initial Tr concentration. This means that Tr has a memory of the mixing history, unlike AF that can be influenced by source and sink processes. A parcel can regain liquid water after a mixing event, if supersaturation is reached again at a later stage. Moreover, the parcels' condensation rate can be different from that predicted by the adiabatic parcel model, because its droplets size distribution have changed and the local profiles of supersaturation can be very different from the ones of the core. This means that a parcel in the margins of the cloud can be diluted, decreasing both Tr and LWC (AF), but later, if the parcel gains vertical velocity and supersaturation, it might condense water in a rate that is larger than in the core. This will compensate for the LWC loss (keeping Tr the same, while increasing AF; i.e. dLWC/dz>dLWCad/dz). The toroidal vortex seems to be a mechanism that drives such conditions. In Fig. 2a we show red regions of AF>Tr which are voxels of relatively strong updrafts and are part of the flow pattern of the toroidal vortex (for an elaborated discussion about the vortex see Zhao and Austin 2005). Using the velocity field, the regions of AF>Tr can be tracked back in time (back-trajectory) to their earlier location, where the toroidal vortex entrains environmental air. Those parcels that mix with entrained air are first diluted, and then flow upward driven by the flow in the toroidal vortex. These diluted parcels with low droplets concentration and high vertical velocity create high supersaturation values (higher than the values in the core for the same altitude). Hence, they condense water in a higher rate, which leads to local increase of AF with altitude. The phenomenon of rapid growth of droplets in an updraft following an entrainment event was suggested as a mechanism for rain initiation (Baker et al., 1980 and Yong et al.,2016). Correlations of the red regions (where AF>Tr) with strong updrafts (as part of the toroidal vortex), high supersaturation values and low droplets concentration were found for different time-steps and different cloud simulations."*

[Figure]

**Figure R2. Cross sections of the upper part of the cloud.** For the time of maximal development of the cloud (33 min.) with CCN concentration of 500 cm$^{-3}$. **(a)** The difference between the normalized concentration of the tracer (Tr) and AF. Green contour marks regions where AF>Tr and magenta contour marks the cloud core where Tr=0.8; **(b)** the supersaturation; **(c)** the droplets concentration $N_d$; **(d)** the absolute value of AF.

4. The authors say that one of the main objectives is to assess the methods used for computing the AF from the data generated from the field campaigns. Can the authors provide additional references to show which method is used for which field campaign and shed some light on how field data could be used for estimating AF? For e.g., what information is available during a field campaign and what calculations are conducted.

**Answer:** Thank you for the comment. In this manuscript we point out that there are many studies that use AF as a measure of mixing. Nevertheless, the details of its calculation from a given data set are usually missing (lines 123-125). We mention some studies that describe their approaches; like Gerber et al. (2008) who used eq. (7) and Schmeissner et al., (2015) who used eq. (8). Most studies calculated LWC$_{ad}$ according to the saturation adjustment assumption as used in plotting a tephigram (Rogers and Yau, 1989; Khain and Pinsky, 2018). Accurate estimation of AF demands knowledge of the humidity and temperature profiles and of cloud base height. Those can be obtained by different ways in field campaigns. We revised the text to describe it shortly (see below). The present

study aims to give an overview of the existing methods to calculate AF, and to analyze and emphasize the limitations of this basic and important variable. We hope that the results of this study can help any researcher (modeler or observer) to consider the options of how to use AF according to the limitation of this variable and the data that is used. This point is emphasized more clearly in the revised introduction and conclusions parts, and examples of current methods to acquired related data in the field are given.

Lines 67-74 in the introduction: *"Accurate estimation of AF demands knowledge of the humidity and temperature profiles and of cloud base height (as shown below in sect. 2.3), which are obtained in various ways in field measurements. While the humidity and temperature profiles can be obtained by radiosondes, aircraft profiling trajectories or remote sensing, the cloud base height can be estimated using calculation of the lifting condensation level (LCL), Lidar/ceilometer measurements or direct sampling according to visual identification from an aircraft. The supersaturation profile, which is a non-linear function of the humidity and temperature profiles, cannot be measured in the field at a suitable precision to the best of our knowledge. The different techniques by which the data was acquired will determine the resolution and precision, thus, affecting the best choice of method to calculate AF. "*

Lines 81-87 in the introduction: *"The simplicity and importance of AF make it applicable in many different data sets of both modeling and measurements. Since every observational data set will have different limitations (or models; e.g. varying schemes and resolutions), it is impossible for this paper to suggest one general solution to all (i.e. one algorithm of AF). This study uses a simple framework of a single cloud, while solving many of the interior complexities that affect AF, to suggest some tools for calculations of AF, and to present the limitations one might encounter while doing so."*

Lines 376-381 in the conclusions: *"This enables a better representation of mixing, and relaxes the dependency on sub-grid parameterization schemes. A sub-cloud layer's passive tracer (Tr), which is an accurate measure of mechanical mixing, was added to the simulations and used as a reference. This model configuration enabled to better control AF and the complex processes that it represents, and to give a theoretical framework that allowed testing the accuracy of different approaches that are commonly used to calculate adiabatic fraction (AF)."*

5. The standard measurement of entrainment/mixing is done via liquid water potential temp. Can't these conserved variables be used for calculating the AF from field measurements?

**Answer:** Variables that are conserved during expansion of air and phase changes such as total water mixing ratio and liquid water potential temperature ($q_t$ or $\theta_l$, respectably) are often used to estimate the mixing level of a cloud. Those variables are similar to some extent to the passive tracer that we used, i.e.

characterize dilution. These conservable variables have an advantage over our theoretical passive tracer because they can be measured in the field. At the same time, as we point out in lines 35-41, the limitation of such variables is that they exist also outside of the cloud and they change with height (they can also be different at the different sides of a cloud). This means that mixing with environmental air acts as another kind of source of $q_t$ or $\theta_l$. This is not the case for a sub-cloud passive tracer because we set it to zero above the cloud base (at the initiation stage). All of the arguments above mean that when using $q_t$ or $\theta_l$ one has to assume some mixing properties. As an example, it is common to assume that a parcel experiences a onetime (immediate) discrete mixing event of cloudy parcel (with values calculated at cloud base) with the environment, and a linear combination of the volumes is assumed, such that:

$$\theta_l(x, y, z) = \theta_l^{base} \chi + \theta_l^{env}(z)(1 - \chi)$$

Where $\chi$ is the cloud volume fraction, $\theta_l^{base}$ is the liquid water potential temperature at cloud base and $\theta_l^{env}(z)$ is for the environment at the level of observation. From this, one can obtain the mixing level as:

$$\chi = \frac{\theta_l(x, y, z) - \theta_l^{env}(z)}{\theta_l^{base} - \theta_l^{env}(z)}$$

Since mixing is a continuous process with relaxation time that can be significant and that a parcel can experience multiple events during its lifetime, these assumptions exert limitations on using natural conservable variables as measures of mixing. For this reason, we focus in our study on AF and its limitations. We believe that this is more applicable in field measurements. Moreover, we are interested largely in the microphysical variables like LWC, which experience evaporation and condensation. The mentioned lines were reformed to be clearer.

Lines 35-41: *"It is common to use conservative variables such as total water mixing ratio or equivalent potential temperature as they can be measured in the field. These variables' limitation is that they exist also outside of the cloud and above its base. This means that using these variables for estimation of the mixing level of cloudy volumes demands knowledge about their environmental profile and assumptions on the mixing processes. Sub-cloud tracer is preferable over these natural variables, as it is absent from the clouds' surroundings. However, such fictitious tracer do not exist in in-situ measurements and remote sensing and is only being used in numerical simulations, aiming for process-level understanding of mixing."*

Note, the LWC$_{ad}$ and AF can be evaluated using conservative variables as total water mixing ratio (see Eq. 7) or conservation of moist static energy (Eq. 8). The problem is that the present methods show higher sensitivity to the choice of "adiabatic profiles" and that they do not allow to measure supersaturation. Accordingly, effects of saturation adjustment and other simplifications cannot be evaluated when using them.

6. One of the key difficulties in estimating the adiabatic LWC from the field data is related to knowing where the cloud base is located. If the location and the condition at the base are known, then plotting a moist adiabat is sufficient to know the properties of the adiabatic parcel (Rogers and Yau 1996, Bohren and Albrecht 2000). In the current study, there is no mention or discussion about this method. In my opinion, this would be the most fundamental method from the point of view of the field data, provided we know the height of the cloud base and its properties. Can the authors comment on this? There might still be issues related to supersaturation that needs to be investigated.

**Answer:** We thank the reviewer for the comment that enabled us to clarify the text. The method described by Rogers and Yau is the first method that is tested and discussed in section 3.2.1 (Linear $LWC_{ad}$). This method assumes that the moist lapse rate of a parcel is constant with height, hence we can deduce $LWC_{ad}$ by knowing the temperature and humidity at cloud base. The manuscript is referring to the short paper of Pontikis (1996) that derived $LWC_{ad}$ (see Eq. 8 in this paper), using the definition of Roger and Yau to the moist adiabatic lapse rate (Eq. 7 in Pontikis 1996 is similar to Eq. 3.16 in Rogers and Yau). We note that the final solution given by Pontikis is similar to our solution of constant ratio of $A_1/A_2$ (see definitions of the parameters in eq. 2a,2b); the difference is with the used units (mixing ratio vs. density), hence in a factor of the density of dry air ($\rho_a$). We added a reference to Roger and Yau in the text and noted that our solution is identical to the solution of Pontikis.

Line 278: *"This implies that $A_1/A_2$ can be used as a constant, based on the known values at the cloud base. Note that the derivation of AF using Rogers and Yau (1996) leads to the equation of $A_1/A_2$ at cloud base."*

7. Line 355: "condensation that occurs after a mixing event can delete records of earlier evaporation/dilution events" – I do not think this statement is supported by the data or the discussion presented in this work. The LWC a parcel attains at a given height is an integrated effect of past entrainment/mixing/evaporation/condensation events. Without knowledge of this history, the final LWC cannot be computed. So, I do not understand the above-quoted statement from the authors. If the authors do not agree, they need to provide strong evidence to support this statement.

**Answer:** This comment is related to comment number 3. A deeper discussion about this point is given in lines 195-200 in the original manuscript and in our answer above.

8. Finally, the title of the paper is a bit too general. The objective of this study appears to be to assess various techniques used for estimating the adiabatic LWC and does not shed much light on the adiabatic core/mixing processes/adiabatic fraction in cumulus clouds. The authors could come up with a more specific title that reflects the scope of the work.

**Answer:** We thank the reviewer for this comment, the title was changed to describe the paper more clearly: *"Revisiting Adiabatic Fraction Estimations in Cumulus Clouds: High-Resolution Simulations with Passive Tracer"*

**Minor Comments:**

1. The abstract should contain the key results/conclusions of this study.

**Answer:** The main results were added to the abstract:

*" Comparison of three different methods to derive AF to the passive tracer show that one method is much more robust than the others. Moreover, this methods' equation's structure also allows to isolate different assumptions that are often practiced when calculating AF such as: vertical profiles, cloud base height, and the linearity of AF with height. The use of a detailed spectral bin microphysics scheme allows accurate description of the supersaturation field and demonstrates that the accuracy of the saturation adjustment assumption depends on aerosol concentration, leading to an underestimation of AF in pristine environments. "*

2. Line 21: Diffusion efficiency of what?

**Answer:** The line was revised for clarity (Line 23): *"As an example, high aerosol loading conditions increase the number of droplets and their surface area to volume ratio, which increases the rates of condensation or evaporation"*

3. Line31: conserved and not "conservative".

**Answer:** Corrected, thank you.

4. Lines 36-38: Can the authors give examples of scenarios when radiation and sedimentation effects can be neglected.

**Answer:** We appreciate this comment. Sedimentation can be neglected when the liquid water droplets total mass is dominated by droplets smaller than ~30 µm, this holds mostly for clouds with high droplets concentration and in the lower parts of growing clouds. The averaged droplet radius of typical marine boundary layer clouds in the trades does not exceed 15 µm, i.e. fall velocity is less than 2-3 cm s$^{-1}$. It means that the vertical shift of droplets with respect to ascending air does not exceed ~10 m (which is the vertical resolution used here) during the growing stage, i.e. negligibly small.

Since clouds strongly reflect the solar radiation, absorption can be neglected; radiative cooling might be significant near the top of clouds below dry atmosphere. An example for such scenario is the cloud top of stratocumulus clouds. In our case, we made sure that our analysis is free of large sedimentation rate (see lines 321-326) and we did not implemented radiation transfer model in the simulation. This is now specifically written in line 175-178:

*"In this work, we analyze the growth and mature stages of shallow cumulus clouds, before obtaining considerable sedimentation flux. Shallow Cu life time in general is short, hence the radiative heating by the weak absorption of solar radiation or cooling by thermal radiation emittance can be neglected. Therefore,*

*we did not calculate radiation transfer during the simulation. Neglecting sedimentation and radiation allows to use AF as a measure of mixing."*

5.  Before Eq 8.: the definition of moist static energy is not correct. h = Lvqv+CpT+gz. Some additional clarification/steps are required in deriving Eq. 8.

**Answer:** We thank the reviewer for this comment. The typo was corrected and the assumption of conservation of water mass was added to explain the translation of $q_v$ to $q_l$:

*"The third approach is to use the conservation of moist static energy (h), where h = $L_w q_v$ + cpT + gz. Differentiating h with respect to z, conserving it with height (dh/dz = 0), assuming water mass conservation (i.e., $dq_v$=-$dq_l$) and multiplying by ρd gives (Schmeissner et al., 2015):"*

---

## Author Comment (AC2)

**Replay to the reviewers**

**"What is adiabatic fraction in cumulus clouds: high-resolution simulations with passive tracer" by Eytan et al. 2021**

We would like to thank the reviewers for their comments that helped us to improve and clarify the manuscript.

To address all the comments and remarks, the manuscript has been revised. We summarize below the novelties and the main modifications performed in the manuscript to clarify and strengthen the results:

1. The significance and applicability of the results are now emphasized and clearly stated.

2. The abstract and introduction were revised to emphasize that this is a theoretical study that compares different approaches that were used previously to calculate AF, and revisits some of their assumptions.

3. The summary and conclusions section was revised such that the take home message for calculating AF (e.g. list of points to consider) are summarized clearly.

4. The results are explained in a more detailed way now with additional information. For example; the effect of the toroidal vortex on the differences between AF and a conservable scalar.

- Point by point responses are presented below (in blue) and changes are marked in yellow in the text of the revised manuscript.

**Reviewer 2:**

Review report of "What is adiabatic fraction in cumulus clouds: high-resolution simulations with passive tracer" by Eytan et al. 2021

**General comments**:

This manuscript compares various method in calculating adiabatic fraction in cumulus clouds, which is an import parameter in quantifying the mixing level. The authors compared three ways in calculating the adiabatic LWC and their impacts on the AF. Besides, the authors examine the assumptions made in previous studies about the calculation of AF. This manuscript is clear about their method but lacks more explanations in the application of this study. I recommend a revision to emphasize the significance of this study more in either introduction or the discussion paragraph. Other specific comments and the minor corrections are below.

**Answer:** We thank the reviewer for this comment. The introduction and conclusion sections were revised to highlight the significance of the study and to state more clearly what practical actions can be deduced from the study when calculating AF. Please see below detailed answers to all the comments.

**Specific comments**:

1) This study mentioned that it aims at identifying the errors from the assumption in calculation of AF in observations, but the manuscript only shows the results from an idealized model study. Can the author provide more clear linkage that how the results can improve the analysis of observation data?

**Answer:** In this manuscript we point out that there are many studies that use AF as a measure of mixing. Nevertheless, because this variable is so basic, the details of its calculation from a given data set are usually missing (lines 123-125). Several in-situ measurements present values of AF in Cu, usually by assuming that it is linear with height and saturation adjustment. Here we offer a theoretical study that aims to give an overview of the existing methods used to calculate AF, and to analyze and emphasize each method's limitations, together with the general limitations of AF. Since the number of studies that use AF is large and includes both modeling and measurements, whom each one has different data sets; we are unable to address all studies and give a unique solution to the problem. We hope that the results of this study can help any researcher (modeler or observer) to consider the options of how to use AF according to his particular data. As an example, we discuss the strong limitation of using AF in deep convective clouds or point out to biases that saturation adjustments can cause under different aerosols conditions. In order to provide a linkage to field measurements we added sentences that describe the data that is acquired in the field. These points is now more emphasized in the introduction and conclusions parts.

Lines 67-74 in the introduction: *"Accurate estimation of AF demands knowledge of the humidity and temperature profiles and of cloud base height (as shown below in sect. 2.3), which are obtained in various ways in field measurements. While the*

*humidity and temperature profiles can be obtained by radiosondes, aircraft profiling trajectories or remote sensing, the cloud base height can be estimated using calculation of the lifting condensation level (LCL), Lidar/ceilometer measurements or direct sampling according to visual identification from an aircraft. The supersaturation profile, which is a non-linear function of the humidity and temperature profiles, cannot be measured in the field at a suitable precision to the best of our knowledge. The different techniques by which the data was acquired will determine the resolution and precision, thus, affecting the best choice of method to calculate AF. "*

Lines 81-85: *"The simplicity and importance of AF make it applicable in many different data sets of both modeling and measurements. Since every observational data set will have different limitations (or models; e.g. varying schemes and resolutions), it is impossible for this paper to suggest one general solution to all (i.e. one algorithm of AF). This study uses a simple framework of a single cloud, while solving many of the interior complexities that affect AF, to suggest some tools for calculations of AF, and to present the limitations one might encounter while doing so."*

2) The reference AF is based on Equation 6 without considering the supersaturation (Line 166). The Section 3.2 shows that not considering supersaturation leads to errors especially in the lower cloud region of cleaner scenario. I think it better to use the full equation (Equation 5) to calculate the LWC_ref since it has less assumption.

**Answer:** We thank the reviewer for this comment. The choice of $AF_{ref}$ is indeed not the choice of the most accurate method, but the choice of the method that can be compared with all other methods. By using $AF_{ref}$ (eq. 6) we can isolate the effect of each assumption or term in the equation. For example, the comparison in fig. 3b only takes $A_1/A_2$ to be constant with height and in fig. 3d only adds the second (supersaturation) term. By comparing the full equation (eq. 5) to the linear assumption (which cannot include profile of supersaturation) we will be mixing biases of both linear and saturation adjustments assumptions.

A line was added to make this point clear (Line 274): *"Each approach will be compared with the reference approach ($AF_{ref}$; Eq. 6). This method is not the most accurate one (method $AF_s$ using Eq. 5 is), but since it is the base for all other examined assumptions, using it allows to isolate and examine each of the assumptions separately.*

3) What are the recommendations for the calculation of AF in the future studies?

**Answer:** We appreciate this direct comment. A short list of the recommendations for all studies that use AF was added to the end of the summary and conclusions section (lines 414-432):

1) Calculations of AF will be most robust when using eq. 5 (or eq. 6 in polluted conditions).
2) When using AF for studies of aerosol-clouds interactions by comparing different parameters conditioned by AF, one cannot make the saturation

adjustment assumption as it underestimates AF in pristine conditions and can bias the results.

3) AF is most sensitive to the definition of cloud base height. Thus, it is important to make sure that the chosen value well represents the investigated cloud or clouds, at the altitude in which most parcels started to condense water.

4) AF in deep convective clouds is prawn to many large errors, and the uncertainty of the calculations is hard to asses (see line 384-391).
   Mainly:

   a) AF is based on a quasi-hydrostatic equation, which is valid for updrafts smaller than 10 m/s.

   b) Supersaturation in clouds with strong updrafts can increase, leading to underestimations of AF if the S term is neglected.

   c) Sedimentation of particles from higher levels of deep clouds can increase LWC in their lower levels and lead to an overestimation of AF.

   d) The rate of change in $LWC_{ad}$ is dominated by the parameter $A_2$, which changes as water vapor is depleted in clouds, meaning that $LWC_{ad}$ ceases to be linear. The large differences expected in deep clouds between the in-cloud and environmental profiles suggest that the latter is prone to large biases when used to predict AF.

4) This study has removed large-scale wind. How are the results influenced by a mean wind?

**Answer:** Adding a mean wind (and shear) can increase entrainment and mixing and affect the cloud evolution and the cloud size. However, the methods of $LWC_{ad}$ (z) calculation will not change. For simplicity we chose to simulate an environment with no large-scale wind. This is now pointed out in lines 83-87:

*"This study uses a simple framework of a single cloud, while solving many of the interior complexities that affect AF, to suggest some tools for calculations of AF, and to present the limitations one might encounter while doing so. External complexities such as advection, wind shear, surface fluxes and variations of aerosols will add complexity to the cloud system, but are not expected to change the nature of AF (only its resulted distribution)."*

**Minor corrections:**
5) Figure 2: it is better to show the differences by the AF-Tr, which is more straightforward for the readers to tell overestimation or underestimation.

**Answer:** Thank you for the comment the figure was changed.

[Figure]

**Figure 2. Cross-sections of the differences between the various AF methods and the tracer.** Vertical cross-sections for the differences between methods using approximated profiles (Fig1e-g) and the tracer (Fig. 1d). (a) Difference between $AF_{ref}$ and Tr (Fig. 1e minus Fig. 1d). (b) Same as a, for $AF_{dtdz}$. (c) Same as a, for $AF_{qt}$.

6) Figure 6 and Figure 7: it is better to use red-white-blue colors to show the differences (similar color scheme as Figure 3b-3f)

**Answer:** We thank the reviewer for the suggestion. Since each panel has a bias of one direction, we prefer to use a colormap that allows a higher resolution in colors and not only different shades of blue or red.

7) Line 224: ql should have the l as subscript?

**Answer:** Thank you, the error was fixed.

---

## Author Response (AR2)

**Replay to Reviewer 1**

**"Revisiting Adiabatic Fraction Estimations in Cumulus Clouds: High-Resolution Simulations with Passive Tracer" by Eytan et al. 2021**

I thank the authors for submitting such a detailed response. Some additional clarification may be required about comment 6.

If the cloud base thermodynamic properties are known then one can obtain the adiabatic parcel temperature and water vapor (assuming saturation adjustment, i.e., thermodynamic equilibrium) at any height. So, why is there a need to assume A1 and A2 to be constant? Even a constant assumption seems very close to AFref (Fig 3b). So if the adiabatic variation in temp. and water vapor is accounted for how will that fair against AFref? All three methods can be re-evaluated using the moist adiabatic parcel temp and water vapor data. My guess is that all three methods will give similar results. It will be nice to see how they compare against AFref?

> Once this comment is addressed satisfactorily, I recommend the acceptance of this manuscript.

**Answer:** We thank the reviewer for this comment and we clarify the regarded points below.

The coupling between the condensate mass in an adiabatic parcel (LWC$_{ad}$) and the temperature profile requires knowledge about one of them for calculating the other. Hence, it does not allow accurate estimation of the LWC$_{ad}$ based only on knowledge of cloud-base temperature and humidity. When taking analytical methods to estimate LWC$_{ad}$ while using only cloud-base properties, some additional assumptions have to be made (e.g., constant moist lapse rate and a constant ratio of A$_1$/A$_2$). This will be explained and demonstrated by the equations below.

In this work we considered only analytical approaches (rather than numerical solutions), since they are more abundant in the literature and because they are simpler to use and are cheaper computationally. Below we derive the analytical solutions according to the book by Rogers and Yau (1996), (which was mentioned by the reviewer in comment no. 6 in the former review), and show that those solutions are identical to some of the methods that were presented and tested in the manuscript.

The change in liquid water mixing ratio (q$_{ad}$) with altitude (which is proportional to LWC$_{ad}$) can be deduced from Rogers and Yau 1996 (page 32, Eq. 3.15). In that equation, one can see that the change in q$_{ad}$ is indeed a function of the temperature profile, as stated above.

Isolating the vertical gradient of water vapor mixing ratio (q$_v$) in the equation, and assuming mass conservation (the decrease in water vapor is due to condensation and increase in liquid water) gives:

$$(1) \qquad \frac{-dq_v}{dz} = \frac{dq_{ad}}{dz} = \frac{c_p}{L}\left(\frac{dT}{dz} - \frac{TR_d}{c_p p}\frac{dp}{dz}\right)$$

where T is the temperature, $c_p$ is the heat capacity of air, p is pressure, L is latent heat and $R_d$ is the gas constant of dry air.

Assuming hydrostatic balance and an ideal gas gives:

$$(2) \qquad \frac{dq_{ad}}{dz} = \frac{c_p}{L}\left(\frac{dT}{dz} + \frac{g}{c_p}\right)$$

where g is the acceleration of gravity.

This solution (Eq. 2 here) is identical to Eq. 8 in the manuscript (method $AF_{dTdz}$; derived from the moist static energy) that is tested and presented in the paper. Note that the difference in a factor of dry air density comes from the usage of different units (mixing ratio in the book and density in our manuscript). As the reviewer mentioned, this method is comparable to the main (reference; $AF_{ref}$) method in the manuscript, but only in ideal conditions; with accurate information on the cloud's core profiles (see Fig. A2 and Fig. 1).

In the manuscript, we tested this method by taking the temperature profile (dT/dz) of the cloud's core. Another approach is to approximate the temperature profile by considering the saturation adjustment assumption and using the Clausius–Clapeyron equation (i.e. referred by the reviewer as the moist adiabat). This is given in Rogers and Yau 1996 by Eq. 3.16:

$$(3) \qquad \frac{dT}{dz} = -\Gamma_d \frac{1 + \frac{Lq_{vs}}{R_d T}}{1 + \frac{L^2 q_{vs}}{R_v c_p T^2}}$$

where $\Gamma_d$ is the dry adiabatic lapse rate and is close to $g/c_p$.

Substituting Eq. (3) into Eq. (2) yields a solution that is identical to Eq. 6 in the manuscript which is the reference method ($AF_{ref}$).

If one wishes to simplify $LWC_{ad}$ calculations and use only the cloud-base properties, it is possible to assume (under some conditions, as shown in the manuscript) a constant moist lapse rate (i.e. using the humidity and temperature values at cloud base). This method ($AF_{linear}$) was applied in the paper by assuming a constant value of the ratio of $A_1$ and $A_2$ above cloud base. It is identical to solving Eq. 3 from above by using the cloud base temperature and water vapor mixing ratio.

Finally, we emphasize again an important point in the manuscript: the derivation and equation that we chose to use in the manuscript (given below as Eq. 4) gives the full equation of $LWC_{ad}$ without the saturation adjustment assumption. This allows to consider the bias of this almost inherent assumption in $LWC_{ad}$ calculations under different conditions.

$$(4) \qquad LWC_{ad}(z) = \int_0^z \frac{A_1(z')}{A_2(z')} dz' - \int_0^z \frac{1}{A_2(z')} \frac{dS}{dz'} dz'$$